# NFκB and NLRP3/NLRC4 inflammasomes regulate differentiation, activation and functional properties of monocytes in response to distinct SARS-CoV-2 proteins

Ilya Tsukalov[1,11], Ildefonso Sánchez-Cerrillo[2,3,11], Olga Rajas[4], Elena Avalos[4], Gorane Iturricastillo[4], Laura Esparcia[1,2], María José Buzón [5], Meritxell Genescà [5], Camila Scagnetti[2], Olga Popova[1], Noa Martin-Cófreces[1,2], Marta Calvet-Mirabent[1,2], Ana Marcos-Jimenez [1,2], Pedro Martínez-Fleta[1,2], Cristina Delgado-Arévalo[2], Ignacio de los Santos[3,6], Cecilia Muñoz-Calleja[2,3], María José Calzada[1,7], Isidoro González Álvaro[8], José Palacios-Calvo[9], Arantzazu Alfranca [2,10], Julio Ancochea [4], Francisco Sánchez-Madrid[1,2,10] & Enrique Martin-Gayo [1,2,3] ✉

Increased recruitment of transitional and non-classical monocytes in the lung during SARS-CoV-2 infection is associated with COVID-19 severity. However, whether specific innate sensors mediate the activation or differentiation of monocytes in response to different SARS-CoV-2 proteins remain poorly characterized. Here, we show that SARS-CoV-2 Spike 1 but not nucleoprotein induce differentiation of monocytes into transitional or non-classical subsets from both peripheral blood and COVID-19 bronchoalveolar lavage samples in a NFκB-dependent manner, but this process does not require inflammasome activation. However, NLRP3 and NLRC4 differentially regulated CD86 expression in monocytes in response to Spike 1 and Nucleoprotein, respectively. Moreover, monocytes exposed to Spike 1 induce significantly higher proportions of Th1 and Th17 CD4 + T cells. In contrast, monocytes exposed to Nucleoprotein reduce the degranulation of CD8 + T cells from severe COVID-19 patients. Our study provides insights in the differential impact of innate sensors in regulating monocytes in response to different SARS-CoV-2 proteins, which might be useful to better understand COVID-19 immunopathology and identify therapeutic targets.

In a small proportion of individuals, SARS-CoV-2 infection can lead to an increased risk to develop severe Coronavirus Disease 2019 (COVID-19)[1], which is characterized by the progression into life threatening conditions, including pneumonia, acute respiratory distress syndrome (ARDS) and cardiovascular disease[2–4]. It has now been established that a number of risk factors including pre-existing inflammatory clinical conditions[5], immunosuppression[6–8] and genetic factors affecting the generation of interferon (IFN)-specific autoantibodies[9,10] can influence the development of severe COVID-19 pathology. Several studies have provided evidence supporting hyperinflammatory and dysregulated immune responses in severe COVID-19 patients. Such inflammatory state is characterized by increased detection of proinflammatory

cytokines in plasma, likely produced by activated or de-regulated myeloid cells such as monocytes (Mo) and dendritic cells (DC)[11,12].

Mo are innate immune cells that differentiate from hematopoietic precursors in the bone marrow[13] and can be divided into three groups: classical (C-Mo), transitional or intermediate (T-Mo) and non-classical (NC-Mo) subsets[14]. C-Mo are the most undifferentiated stage and express broad range of genes related to microbial innate sensing, proinflammatory molecules, participate in phagocytosis and are able to differentiate into CD14+ interstitial DC[15,16]. T-Mo present high phagocytosis potential, high ROS production and high MHC-II expression[14,17]. NC-Mo are involved in complement, FcR-mediated phagocytosis and are associated with the clearance of cell debris and antiviral immune responses[18]. T-Mo and NC-Mo subsets can be recruited to inflamed tissues[14,19,20] and their proportions are increased during bacterial and viral infection and in inflammatory diseases[16]. In fact, our group and others identified the reduction in the proportions of specific subsets of myeloid cells such as Mo, DC and neutrophils in the peripheral blood (PB) and their enrichment in bronchial infiltrates from severe COVID-19 patients, suggesting that these cell subsets play a key role in the immunopathogenesis of this disease[21,22]. Also, T-Mo and NC-Mo enriched in lung infiltrates of critical COVID-19 patients are characterized by an increased activation state[21]. However, the influence of different SARS-CoV-2 proteins in these specific Mo subsets and the mechanisms driving their differentiation, activation and their potential functional ability to mediate T cell activation is not well understood yet in COVID-19 patients.

Two main molecular pathways have been proposed to play a role in the innate response of Mo to SARS-CoV-2: Toll-like Receptors (TLR) signaling and the inflammasome. At the steady state, Mo basally express high levels of TLR1, 2, 4, and 5[23]. TLR4 and TLR2 are expressed at higher levels in C-Mo[24] but are also present in T-Mo and NC-Mo subsets. It has been shown that SARS-CoV-2 Spike (S) protein may be able to activate TLR2 and/or TLR4 in Mo[25–28]. This suggests that different Mo subsets may differ in their ability to detect and respond to SARS-CoV-2 components. However, whether the response of Mo to SARS-CoV-2 involves the activation of other alternative TLR pathways associated with viral or microbial infection such as TLR3 and TLR5 has not been investigated.

The inflammasome is an intracellular multiprotein complex which can be assembled by the recruitment of a variety of NOD-like Receptors (NLR) Upon formation, it activates caspase-1, leading to pro-IL-1β and pro-IL-18 cleavage and secretion of their active forms. Inflammasome signaling plays an important role in immune defense, but it can also lead to dysregulation in autoinflammatory diseases, infections, and cancer[29,30]. Different inflammasome sensors are triggered in response to pathogen and damage associated molecular patterns (PAMPs and DAMPs, respectively), metabolites, potassium, and intracellular nucleic acids[31,32]. The NLRP3 inflammasome can be induced downstream TLR signaling[33] and has also been involved in the innate sensing of viruses, including SARS-CoV-2 S protein[28]. However, the potential role of other inflammasomes in the sensing of SARS-CoV-2 in Mo remains understudied. In this regard, the NLRC4 inflammasome also contributes to TLR activation and participates both in antimicrobial[34] and autoimmune inflammatory responses[35].

Hyperinflammation in the lung during SARS-CoV-2 infection has been linked to dysfunctional and exhausted SARS-CoV-2-specific CD4+ and CD8 + T cells, which play an important role in viral clearance and control of SARS-CoV-2[36–38]. CD4 + T cells can be found in the lung infiltrates of COVID-19 patients and have been reported to express a variety of cytokines including IFN-γ[39,40], IL-17[41], Granulocyte-macrophage colony-stimulating factor (GM-CSF)[42] and IL-4[43] in response to antigenic stimulation, suggesting that different CD4 + T cell subsets might participate in the inflammation and pathogenesis of COVID-19[44]. On the other hand, previous studies also suggest that CD8 + T cell from severe COVID-19 patients possess a basal hyperactivated and increased cytotoxicity state compared to healthy individuals[45]. In line with these observations, we previously showed that higher proportions of T-Mo correlate with higher detection of hyperactivated CD38 + CD8 + T cells in the lung from severe COVID-19 patients[21]. However, the functional properties of hyperactivated CD8 + T cells, how they become activated in the lung of COVID-19 patients and which SARS-CoV-2 proteins might trigger these responses remain to be determined. Interestingly, cytotoxic CD107a + CD8 + T cells seem to be differentially induced in response to nucleoprotein (NP) peptides in mild and severe COVID-19 patients[46]. Additional studies report that CD8 + T cells from COVID-19 patients are characterized by lower basal levels of expression of IFN-γ, IL-2, and CD107a and this defect may be associated with immune exhaustion[47]. Therefore, the impact of SARS-CoV-2 proteins on CD8 + T cell responses remains unclear. Also, tissue-resident CD103 + CD8 + T cells may play an important role during COVID-19 pathogenesis in the lung[46] and might display different profiles from peripheral CD103- T cells[48], which are also recruited to this tissue in severe COVID-19 patients[49]. In addition, how innate sensing pathways associate with functional properties of specific Mo subsets that might affect T cell responses during COVID-19 has not been addressed. This study is focused on better understanding whether different SARS-CoV-2 proteins may modulate the differentiation and activation of distinct Mo subsets and contribute to their activation through specific innate sensing pathways. We have shown that S1 and NP are both able to induce inflammasome activation. In addition, S1 is able to promote differentiation of T-Mo and NC-Mo in a NFκB-dependent manner. Also, we assessed whether the recognition of different SARS-CoV-2 proteins by Mo influence their functional ability to modulate specific CD4+ and CD8 + T cell subsets. Our data indicate that NP- and S1-primed Mo may contribute to reduced cytotoxic CD8 + T cell and increased IFNγ+ and IL-17 + CD4 + T cells in COVID-19 patients, respectively.

## Results

### SARS-CoV-2 S1 and NP proteins differentially induce differentiation and activation of T-Mo and NC-Mo subsets

To evaluate the impact of different SARS-CoV-2 proteins on the generation and activation of different Mo subsets, we used PBMC from our cohort of healthy donors and critical COVID-19 patients (Supplementary Table 1) and stimulated them with pools of peptides from region 1 of S (S1) and NP SARS-CoV-2 proteins or with ligands for TLR associated with viral (Poly I:C, TLR3 ligand) or bacterial (flagellin, TLR5 ligand) infection. Subsequently, we defined proportions of CD14hi CD16- classical (C-Mo), CD14hi CD16+ transitional (T-Mo) and CD14lo CD16+ non-classical (NC-Mo) Mo present in these culture conditions (Supplementary Fig. 1A). In the presence of SARS-CoV-2 S1 peptides, proportions of the NC-Mo and C-Mo subsets present in bulk cultures of PBMCs from healthy donors or critical COVID-19 patients were significantly increased and decreased, respectively (Fig. 1A, Supplementary Fig. 1B). This effect was not observed in the presence of SARS-CoV-2 NP or Poly I:C. Similar findings in NC-Mo were observed in bulk cultures of bronchoalveolar lavage lung infiltrate cells (BAL) from critical COVID-19 patients (Supplementary Table 2), and in control non-COVID-19 patients (Supplementary Table 3) (Fig. 1A, lower panels). The frequencies of T-Mo but not C-Mo subset were significantly changed in non-COVID19 control samples in contrast to the other healthy and COVID-19 samples after stimulation with any of the SARS-CoV-2 proteins in these assays performed in bulk (Supplementary Fig. 1B). To confirm that changes in NC-Mo and C-Mo proportions in the presence of S1 may underscore an active process of differentiation of Mo in vitro, these experiments were repeated using preisolated Mo from PB of healthy donors. In this case, we also observed a significant increase in proportions of both NC-Mo and T-Mo accompanied by a significant decrease in C-Mo upon S1 stimulation, which did not occur in the presence of NP or Poly I:C (Fig. 1B). Such increase in proportions

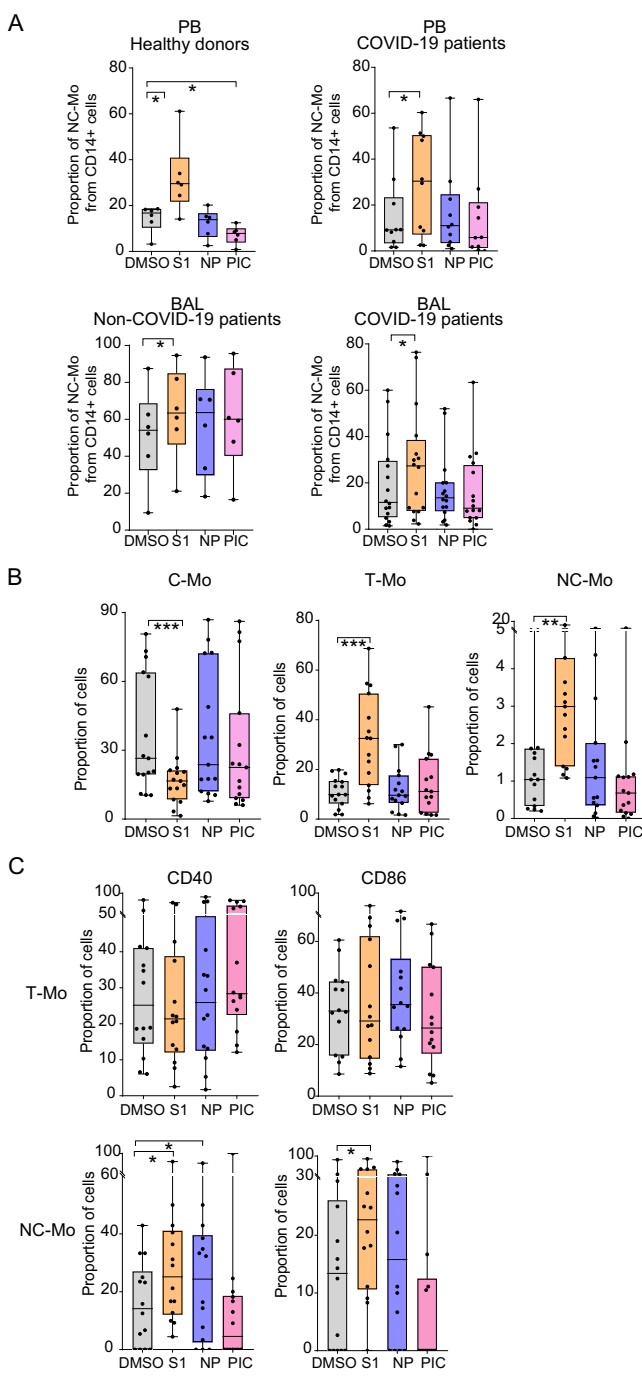

**Fig. 1 | Impact of SARS-CoV-2 S1 and NP on Mo subset differentiation and activation. A**, **B** Proportions of Non-classical (NC-Mo, **A**–**C**), Transitional (T-Mo; **B**, **C**) and classical (C-Mo, **B**) Mo in bulk cultures (**A**) of PBMC of healthy donors (*n* = 6) (left upper **A**) (*p* = 0.0313; 0.0313), COVID-19 patients (*n* = 10) (right upper **A**) (*p* = 0.0322) and of lung infiltrates (BAL) from control non-COVID-19 (*n* = 6) (lower left **A**) (*p* = 0.0313) and severe COVID-19 patients (*n* = 16) (lower right **A**) (*p* = 0.0110) or in pre-isolated Mo (*n* = 15) (**B**) cultured in the presence of DMSO (gray), S1 peptide (orange), NP peptide (purple) and Poly I:C (PIC, pink) (*p* = 0.0001; 0.0001; 0.0042). **C** proportions of CD40+ (left) and CD86+ (right) cells included in the T-Mo subset (upper plots; *n* = 15) and in the NC-Mo subset (lower plots; *n* = 14) from each culture condition (*p* = 0.0200; 0.0322; 0.0479). Data are represented as box and whiskers with bars representing maximum and minimum values and with median highlighted as a line. Statistical significance was calculated using a one-tailed (**A** upper right) or two-tailed (rest of plots) Wilcoxon tests: *$p < 0.05$, **$p < 0.01$, ***$p < 0.001$.

of T-Mo, but not in NC-Mo, was also observed in cells stimulated with flagellin (Supplementary Fig. 1C).

We next evaluated whether, regardless of the observed Mo differentiation, SARS-CoV-2 S1 and NP proteins also induced the activation of these Mo subsets. As previously reported, T-Mo exhibited the highest basal levels of CD40 compared to C-Mo and NC-Mo in the DMSO condition, suggesting that this subset intrinsically represents the most activated state (Supplementary Fig. 1D). In contrast, expression of CD40 and CD86 was markedly lower in the NC-Mo subset at baseline compared to T-Mo and C-Mo (Supplementary Fig. 1D). Upon stimulation with SARS-CoV-2 NP, levels of expression of CD40 did not significantly change in T-Mo but CD86 tended to increase in this subset cultured in the presence of this viral protein compared to the control DMSO condition (Fig. 1C). No obvious effect of S1 on maturation of T-Mo was observed (Fig. 1C). In addition, expression of both CD40 and CD86 were significantly increased on NC-Mo subset in response to SARS-CoV-2 S1 while treatment with NP only increased CD40 expression in this cell subset (Fig. 1C). In contrast, flagellin also increased expression of CD40 on NC-Mo but did not significantly affect maturation of T-Mo (Supplementary Fig. 1E). Together, these results suggest that S1 and NP are both able to mediate activation of Mo but differ in their ability to induce differentiation into NC-Mo and T-Mo subsets.

## Differential activation of NFκB and inflammasome in Mo exposed to SARS-CoV-2 S1 and NP

We next asked whether the previously observed effects on Mo differentiation and activation induced by SARS-CoV-2 S1 and NP could be mediated by redundant or independent innate sensing pathways. In these experiments, Poly I:C and flagellin were included as two control ligands known to induce both inflammasome[50,51] and NFκB activation[52,53]. The two assayed SARS-CoV-2 proteins induced the transcriptional expression of proinflammatory cytokines linked to TLR signaling such as IL-1β and IL-8, which was more significant in the case of S1 in isolated Mo (Fig. 2A, Supplementary Fig. 2A). Similar non-significant trends were observed for IL-6, while levels of TNFα were significantly induced by NP (Supplementary Fig. 2A). In contrast, the levels of CCL3 and IFN-β transcripts were not significantly altered in response to viral proteins (Supplementary Fig. 2A). As expected, transcription of these cytokines was also induced in the presence of control TLR ligands, except for IFN-β which was not significantly induced by Poly I:C in freshly isolated Mo, as previously described[54] (Fig. 2A, Supplementary Fig. 2A). Transcription of the IFN-dependent cytokine IL-18[55] was significantly decreased in Mo exposed to viral proteins (Supplementary Fig. 2A). Consistent with higher levels of TLR activation through NFκB, we observed higher levels of phosphorylation of the p65 in Mo exposed to S1, compared to NP (Fig. 2B). In contrast, Poly I:C and flagellin were able to induce similar levels of phosphorylated p65 in Mo (Fig. 2B). As the inflammasome has been involved in Mo response to SARS-CoV-2, we consistently observed upregulation of the NLRP3 inflammasome sensor in response to viral proteins and Poly I:C, but only significantly in response to NP (Supplementary Fig. 2B). In contrast, we did not observe any significant induction of NLRC4 transcripts in response to viral proteins or TLR ligands (Supplementary Fig. 2B). To evaluate activation of the complex, we analyzed the processing of procaspase1 into active cleaved caspase1 in Mo exposed to S1, NP and TLR ligands. Accordingly, S1 and NP spontaneously induced significant processing of caspase1, suggesting that both proteins were able to induce inflammasome activity (Fig. 2C). Interestingly, S1 tended to be most efficient tested stimuli inducing active caspase 1 in Mo (Fig. 2C). In addition, a protein involved in inflammasome activity downstream Caspase-1 such as Gasdermin D, was also detected and while cleavage of the protein was more variable, it tended to be higher in Mo exposed to

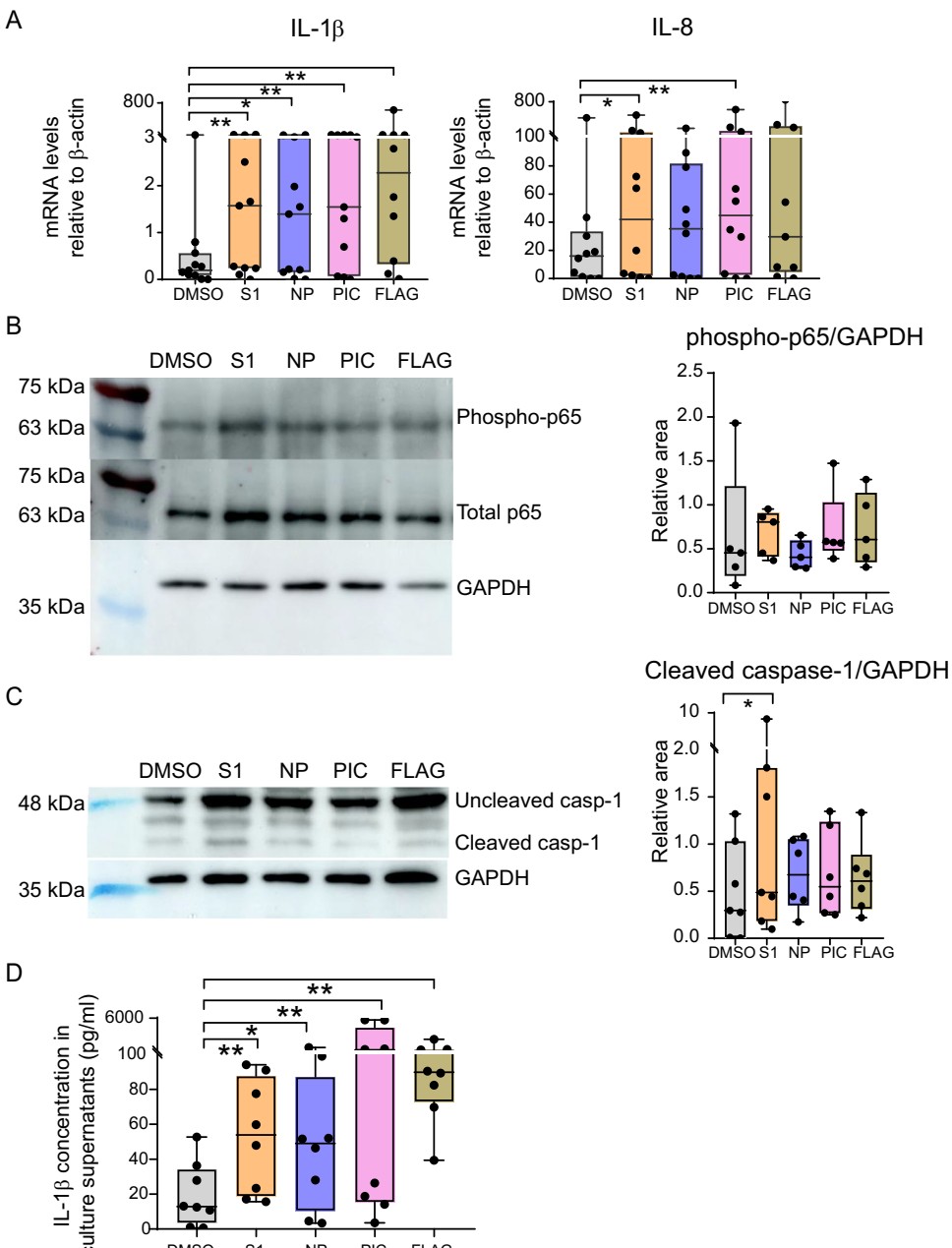

**Fig. 2 | Analysis of proinflammatory cytokine expression and activation of NFκB and Caspase 1 pathways in Mo exposed to SARS-CoV-2 proteins and TLR ligands.** Analysis of mRNA levels (**A**) quantified by RT-qPCR of IL-1β ($n = 11$) (left) ($p = 0.0049$; 0.0420; 0.0049; 0.0059) and IL-8 ($n = 10$) (right) ($p = 0.0371$; 0.0098) normalized to β-actin mRNA expression and (**D**) ELISA quantification of concentration (pg/mL) of secreted active IL-1β protein in culture supernatants of Mo isolated from healthy donors stimulated with DMSO or S1 peptide (orange), NP peptide (purple), Poly I:C (PIC, pink) and Flagellin (FLAG, khaki) ($n = 8$) ($p = 0.0078$; 0.0391; 0.0078; 0.0078). Western blot analysis of phosphorylated and total NFκB p65 subunit (**B**) and uncleaved versus cleaved caspase 1 (**C**) in Mo cultured in the presence of DMSO and 3 h (p65) or 16 h (caspase 1) after stimulation with SARS-CoV-2 S1, NP peptides or TLR ligands. GAPDH was included as a loading control and used for normalization purposes. Quantification of phosphorylated p65 (B; $n = 5$) and cleaved caspase 1 (C; $n = 7$) ($p = 0.0313$) normalized to GAPDH are shown on the right of each panel. Statistical analyses in **A**, **C**, **D** were performed using a two-tailed Wilcoxon test: *$p < 0.05$, **$p < 0.01$. Data are represented as box and whiskers with bars representing maximum and minimum values and with median highlighted as a line.

S1, and Poly I:C (Supplementary Fig. 2C). In line with these observations, stimulation of Mo with S1, NP and TLR ligands significantly increased levels of intracellular IL-1β in these cells (Supplementary Fig. 2D). These findings were associated with a mild but significant decrease in cell viability after exposure to S1 protein (Median 67.6% versus 51.60% of viable cells for DMSO and S1, respectively) (Supplementary Fig. 2E). To distinguish whether pro-IL1β was cleaved into an active isoform, we performed western blot analysis of this protein and observed that levels of both pro-IL-1β and processed IL-

1β increased after stimulation of Mo with SARS-CoV-2 proteins and TLR ligands (Supplementary Fig. 2F). Finally, we corroborated increased levels of secreted active IL-1β detected in culture supernatants compared to the DMSO control condition, confirming an inflammasome activation (Fig. 2D).

To further determine whether TLR and/or inflammasome were differentially involved in the Mo differentiation and activation processes observed in response to SARS-CoV-2 S1, we stimulated isolated Mo from healthy donors with S1 protein in the presence of

pharmacological inhibitors with different levels of specificity against inflammasome and NFκB: parthenolide, a drug with a broad effect which can inhibit both the activity of NFκB (mediator of TLR activity) and caspase-1 (mediator of inflammasome activity)[56,57], a pan-caspase Z-VAD-FMK inhibitor[58] and finally, MCC950 which is a specific inhibitor of the NLRP3 inflammasome[59]. All tested inhibitors significantly reduced concentrations of IL-1β in culture supernatants of Mo exposed

to S1, indicating they efficiently inhibited inflammasomes activity (Supplementary Fig. 3A). In these experiments, we observed that parthenolide significantly abrogated the increase in proportions of T-Mo and NC-Mo after stimulation with S1 peptides (Fig. 3A), while inhibitors of caspases or NLRP3 inflammasome did not exert any significant effect on Mo differentiation (Fig. 3A). Importantly, a similar abrogation of the increase of NC-Mo proportions was observed in bulk cultures of

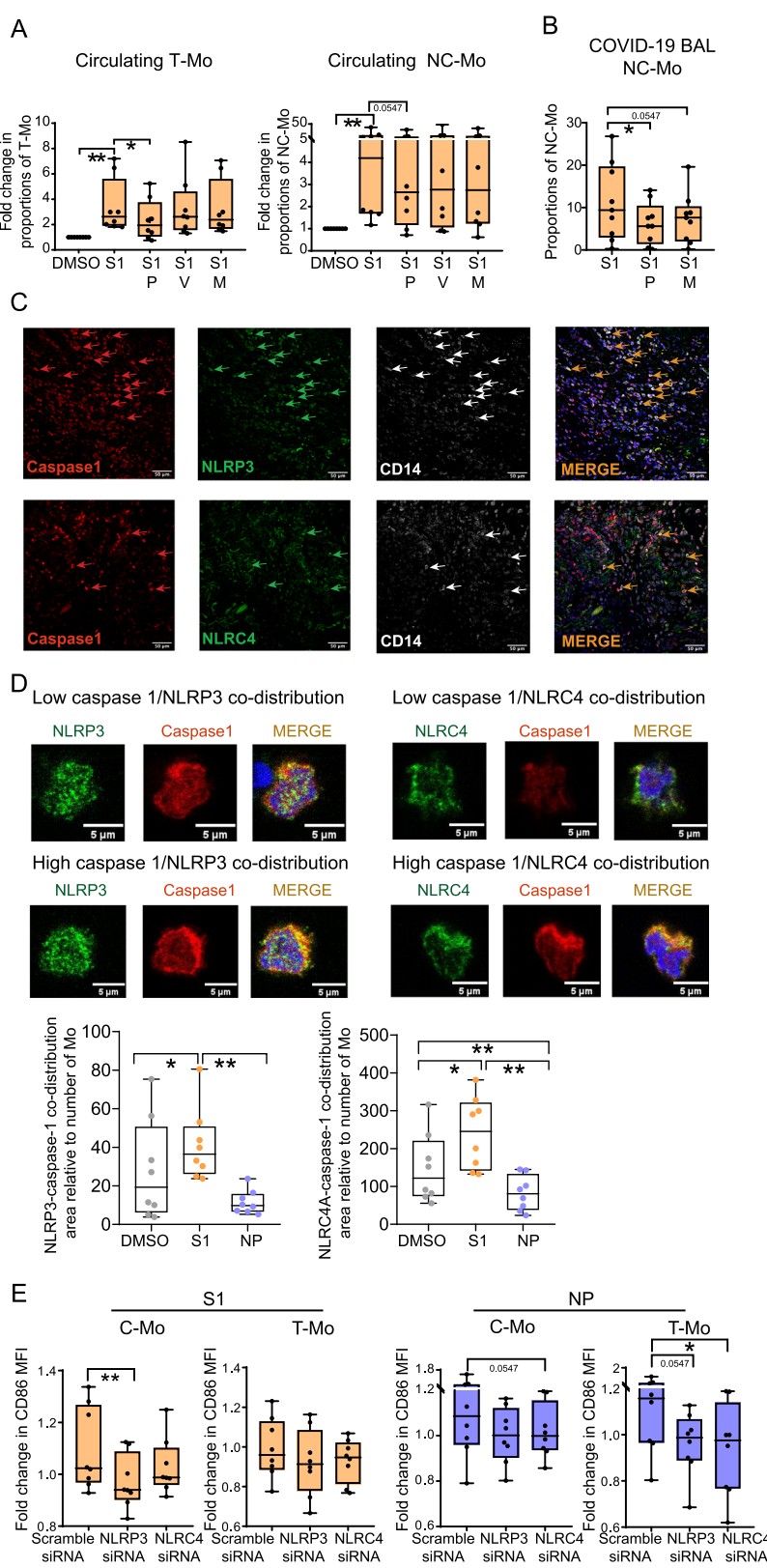

**Fig. 3 | Impact of NLRP3 and NLRC4 inflammasome sensors in Mo response to SARS-CoV-2 S1 and NP proteins. A**, **B** Fold change in proportions (A) or raw proportions (B) of transitional (T-Mo) ($p = 0.0078$; $0.0156$) and non-classical (NC-Mo) Mo ($p = 0.0078$; $0.0547$) subsets included in cultures of pre-isolated CD14+ cells from the blood of healthy donors PB ($n = 8$) (**A**) and in bulk BAL cultures from COVID-19 patients ($n = 9$) (**B**) ($p = 0.0195$; $0.0547$) after stimulation with S1 peptide (S1) in the absence or the presence of Parthenolide (P), Z-VAD-FMK (V) and MCC950 (M). Data were normalized to baseline levels present in control DMSO conditions. For BAL samples only Parthenolide (P) and MCC950 (M) inhibitors were used. **C** Representative 40X magnification confocal microscopy images showing analysis of expression of Caspase-1 (red), NLRP3 or NLRC4 (green, upper and lower panels, respectively), CD14 (white) and merged images including DAPI (orange) from lung tissue from $n = 3$ critical COVID-19 patients. **D** Zoom of representative individual isolated Mo showing different low (upper images) versus high (lower images) co- distribution patterns between caspase-1 (red) and either NLRP3 or NLRC4 (green, left and right panels, respectively). Quantification of co-distribution areas per field of NLRP3 (left) ($p = 0.0234$; $0.0078$) or NLRC4 (right) ($p = 0.0156$; $0.0078$; $0.0078$) with caspase-1 normalized to number of Mo in each field in the presence of DMSO (gray), S1 (orange) or NP (purple) proteins ($n = 3$ patients) are shown below. **E** Impact of siRNA specific for NLRP3 and NLRC4 inflammasome sensors in the fold change of mean fluorescence intensity (MFI) of CD86 expression of Mo from healthy donors Mo exposed to SARS-CoV-2 S1 (left plots) ($p = 0.0078$) and NP (right plots) ($p = 0.0547$; $0.0547$; $0.0156$) proteins in classical (C-Mo, left) and transitional (T-Mo, right) in these assays ($n = 8$). Data are represented as box and whiskers with bars representing maximum and minimum values and with median highlighted as a line. Statistical significance was calculated using a two-tailed Wilcoxon test: *$p < 0.05$, **$p < 0.01$.

BAL from critically infected COVID-19 patients stimulated with S1 (Fig. 3B). As expected, the inhibitory effect of parthenolide was also accompanied by a marked significant decrease in the IL-1β, IL-8, CCL3 mRNA levels in cells stimulated with S1, confirming a reduction of NFκB activity (Supplementary Fig. 3B). Therefore, these data indicate that Mo differentiation mediated by SARS-CoV-2 S1 depends on NFκB-activation.

We next determined which inflammasome sensors might be responsible for the activation of Mo in response to SARS-CoV-2 S1 and NP proteins. We focused on investigating NLRP3, previously involved in innate immune response to SARS-CoV-2[60,61] and NLRC4 which is a sensor with an emerging role mediating pathogenic inflammation[35,62], and may potentially contribute to COVID-19. Notably, high expression of caspase-1, NLRP3 and NLRC4 was found in infiltrated CD14+ Mo present in the lung from critical COVID-19 patients (Fig. 3C). Ratios from CD14+ cells of NLRP3 + CD14+ or NLRP3+ caspase1+ cells tended to be higher than NLRC4+CD14+ or NLRC4+ caspase1+ cells in these tissues (Supplementary Fig. 3C). Moreover, we observed that NLRP3 and both NLRC4 significantly co-distributed with Caspase-1 in purified Mo exposed to SARS-CoV-2 S1 (Fig. 3D). In contrast, no significant increase of co-distribution these sensors and caspase-1 was observed in response to NP (Fig. 3D). These data suggest that NLRP3 and NLRC4 may differentially participate in the innate activation of Mo in response to different SARS-CoV-2 proteins. To further test this possibility, we used siRNA to significantly knock down mRNA expression of NLRP3 and NLRC4 on primary Mo isolated from healthy donors PBMC. siRNA mediated silencing of NLRP3 and NLRC4 mRNA was effective (Min-Max 70–90%) (Supplementary Fig. 3D). Consistent with our previous results, inhibition of NLRP3 and NLRC4 expression did not affect the ability of Mo to differentiate into T-Mo or NC-Mo subsets in response to S1 (Supplementary Fig. 3E). Moreover, we observed differential effects of siRNA-mediated knock down of these sensors in the activation of the Mo subsets in response to either S1 or NP. As shown in Fig. 3E, we observed that NLRP3 silencing significantly decreased the levels of CD86 in C-Mo subset upon S1 stimulation. Similar tendencies of decreased levels of CD86 in T-Mo were observed upon specific NLRP3 silencing and after S1 stimulation (Fig. 3E, right). Interestingly, expression of CD86 in C-Mo and T-Mo subsets upon NP stimulation was significantly more dependent on NLRC4 than on NLRP3 expression, which only partially contributed to this response (Fig. 3E). In the case of NC-Mo, siRNA-mediated knockdown of NLRP3 or NLRC4 did not significantly affect CD86 expression (Supplementary Fig. 3F). In line with these observations, a tendency to lower secretion of IL-1β in culture supernatants after stimulation with S1 and NP proteins was observed in Mo treated with NLRP3 and NLRC4-specific siRNAs, respectively (Supplementary Fig. 3G). Therefore, different inflammasome-inducing sensors differentially induce the activation of distinct Mo subsets, independently of the differentiation from C-Mo.

## Mo exposed to SARS-CoV-2 NP reduce cytotoxic CD8 + T cell responses

We assessed whether stimulation of Mo with SARS-CoV-2 S1 and NP proteins is associated with differential functional abilities to induce T cell activation or promote polarization patterns present in severe and critical COVID-19 patients. We performed in vitro functional assays in mixed lymphocyte reactions (MLR) with Mo from healthy donors exposed to either SARS-CoV-2 S1 or NP peptides or to DMSO, in the presence of allogeneic CD8 + T cells. We then compared these T cell activation patterns with those SARS-CoV-2 specific CD8 + T cell responses observed in PBMCs and BAL from severe COVID-19 patients (Supplementary Tables 1 and 2). We previously reported that higher levels of CD38Hi CD8 + T cells positively correlated with activation of infiltrated Mo[21]. Therefore, we first focused on analyzing CD8 + T cell activation by IFNγ secretion and accumulation of CD107a, as a marker for cytotoxic cells in MLRs and in COVID-19 PBMC and BAL samples (Supplementary Fig. 4A, B). Levels of CD107a were basally significantly increased in CD8 + T cells from severe COVID-19 BAL and PBMCs compared to PBMC from non-COVID-19 controls, and the same trend was observed for IFNγ (Supplementary Fig. 4C). However, we detected that proportions of CD107a + CD8 + T cells were very significantly decreased in the presence of SARS-CoV-2 NP peptides in PBMC from COVID-19 patients, and the same trend was observed in BAL samples ($p = 0.07$) or after culture with Mo exposed to this viral protein ($p = 0.08$) in contrast to myeloid cells exposed to Poly I:C or flagellin (Fig. 4A; Supplementary Fig. 4C, D). Interestingly, expression of CD107a and CD38 on CD8 + T cells followed opposite patterns in MLR and in COVID-19 PBMC and BAL samples (Supplementary Fig. 4E, F). In fact, significantly lower levels of CD107a at baseline were also found in CD38Hi versus CD38 Low CD8 + T cells from COVID-19 BAL samples, and the same tendency was observed after NP stimulation (Supplementary Fig. 4G, left). On the other hand, similar tendencies of decrease of CD107a expression in response to NP was found when analyzing tissue resident CD103 + CD8 + T cells compared to CD103-CD8 + T cells recruited from PB in COVID-19 BAL samples (Supplementary Fig. 4H, left). Our findings suggest that exposure of CD8 + T cells to NP leads to lower proportions of CD107a+ cells and that Mo might participate in this process. In terms of IFNγ expression, we observed that CD38Hi CD8 + T cells present in COVID-19 BAL tended to secrete lower levels of IFNγ than CD38low CD8 + T cells basally and in response to NP stimulation (Supplementary Fig. 4G, right). In addition, tissue resident CD103 + CD8 + T cells tended to produce higher levels of IFNγ upon stimulation with S1 and NP peptides than peripheral CD103- CD8 + T cells (Supplementary Fig. 4H, right). On the other hand, stimulation of PBMC from COVID-19 with S1 induced a significant increase in the proportion of IFNγ + CD8 + T cells, which was not observed in the MLR assays performed with S1-stimulated Mo (Fig. 4B; Supplementary Fig. 4C, D), suggesting that priming of CD8 + T cells in response to S1 may require the presence of other cell types. In contrast, Mo primed with NP induced significantly lower levels of

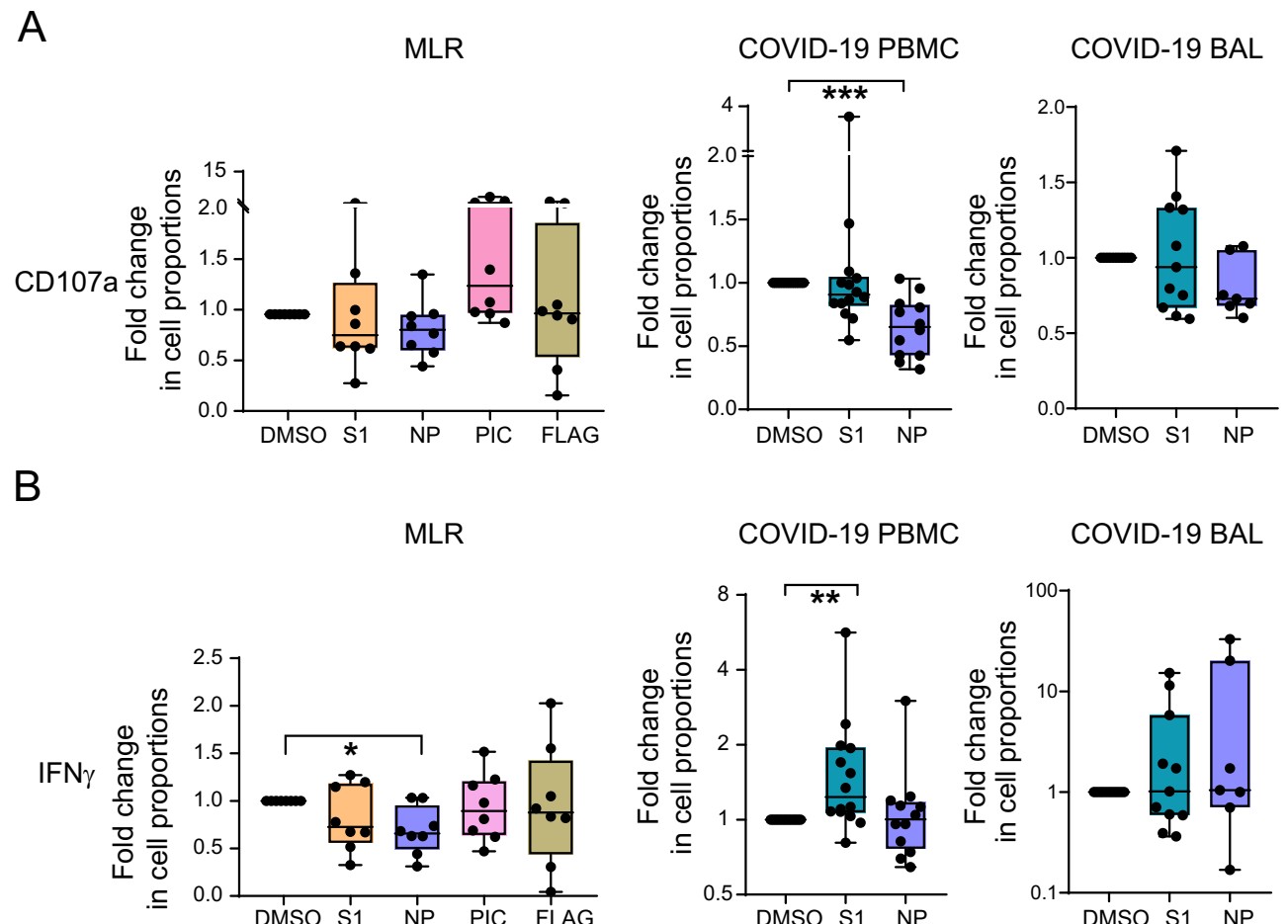

**Fig. 4 | Functional ability of Mo primed with SARS-CoV-2 proteins to activate CD8 + T cells in vitro and comparison with specific CD8 + T cell responses in critical COVID-19 patients.** Fold change in proportions of CD8 + T cells expressing CD107a (**A**) (*p* = 0.0010) and IFNγ (**B**) (*p* = 0.0391; 0.0031) either after 3 days of culture with allogeneic Mo (left panels) primed with DMSO or S1 (orange), NP (purple), Poly I:C (PIC; pink), Flagellin (FLAG, khaki) (*n* = 8) or induced directly by S1 (green) or NP (purple) peptide stimulation in PB (*n* = 14) (middle panel) and BAL (*n* = 11) (right panel) of severe COVID-19 patients. Data are represented as box and whiskers with bars representing maximum and minimum values and with median highlighted as a line. Statistical significance was calculated using a two-tailed Wilcoxon test: \**p* < 0.05, \*\**p* < 0.01, \*\*\**p* < 0.001.

IFNγ + CD8 + T cells in MLR assays (Fig. 4B; Supplementary Fig. 4D). Therefore, Mo exposed to NP but not S1 may more actively influence SARS-CoV-2 specific CD8 + T cell responses in COVID-19 patients.

## Mo exposed to SARS-CoV-2 S1 modulate activation of IFNγ+ and IL-17 + CD4 + T lymphocytes

We additionally assessed whether exposure of Mo to different SARS-CoV-2 proteins could also differentially affect their ability to activate CD4 + T cells. To this end, we focused on analyzing the expression of IFNγ, IL-17, and IL-4 in MLRs performed with Mo exposed to viral proteins co-cultured with allogeneic T cells, or in COVID-19 PBMC, BAL at baseline and after SARS-CoV-2 peptide stimulation (Supplementary Fig. 5A, B). These markers were used as a readout for Th1, Th17, and Th2 cells, which have been reported to be induced during SARS-CoV-2 infection and vaccination[41,43,63,64]. In fact, we observed that basal expression IL-17 and IL-4 was significantly higher in CD4 + T cells from COVID-19 BAL and in PBMC from these patients (Supplementary Fig. 5C). Basal IFNγ expression in BAL samples followed a similar pattern but no significant differences were observed (Supplementary Fig. 5C). As shown in Fig. 5A, Mo primed with S1 significantly increased the proportions of Th1 cells defined as IFNγ + IL-17- CD4 + T in MLRs, which also matched with a significant increase of these cells in COVID-19 PBMC and BAL samples stimulated with S1 peptides (Fig. 5A, Supplementary Fig. 5D). NP also seemed to induce IFNγ responses in BAL

and to some extent in MLR assays (Fig. 5A). Interestingly, a portion of IFNγ+ cells induced in the presence of S1-primed Mo in MLR assays cells also tended to co-expressed high levels of IL-4, suggestive of a pathogenic phenotype (Supplementary Fig. 5E). However, despite high basal levels of IL-4 in BAL samples, we did not observe a consistent increase of IL-4+ cells upon SARS-CoV-2 S1 and NP peptide stimulation (Supplementary Fig. 5C). In contrast, we observed a significant increase in the proportion of Th17 cells defined as IL-17 + IFNγ- CD4 + T cells induced by Mo primed with S1 but not NP in MLR assays (Fig. 5B). In contrast, expression of IL-17 increased in CD4 + T cells from 40% of BAL samples (Fig. 5B). No induction of Th17 cells was observed in CD4 + T cells PB from COVID-19 patients stimulated with S1 (Fig. 5B). Therefore, Mo primed with S1 appear to be more effective inducing polarization of pathogenic-like Th1 and Th17 CD4 + T cells. Together, priming of Mo by SARS-CoV-2 S1 modify their functional abilities to modulate CD4 + T cells that mimic the phenotypes observed in tissues relevant for COVID-19 pathology.

## Presence of NC Mo in the lung of COVID-19 patients associates with severity and the activation of IL-17 + CD4 + T cells in response to S1

We finally assessed whether there was an association between the impact of different SARS-CoV-2 proteins in the differentiation and functional specialization of Mo observed in vitro and the presence of

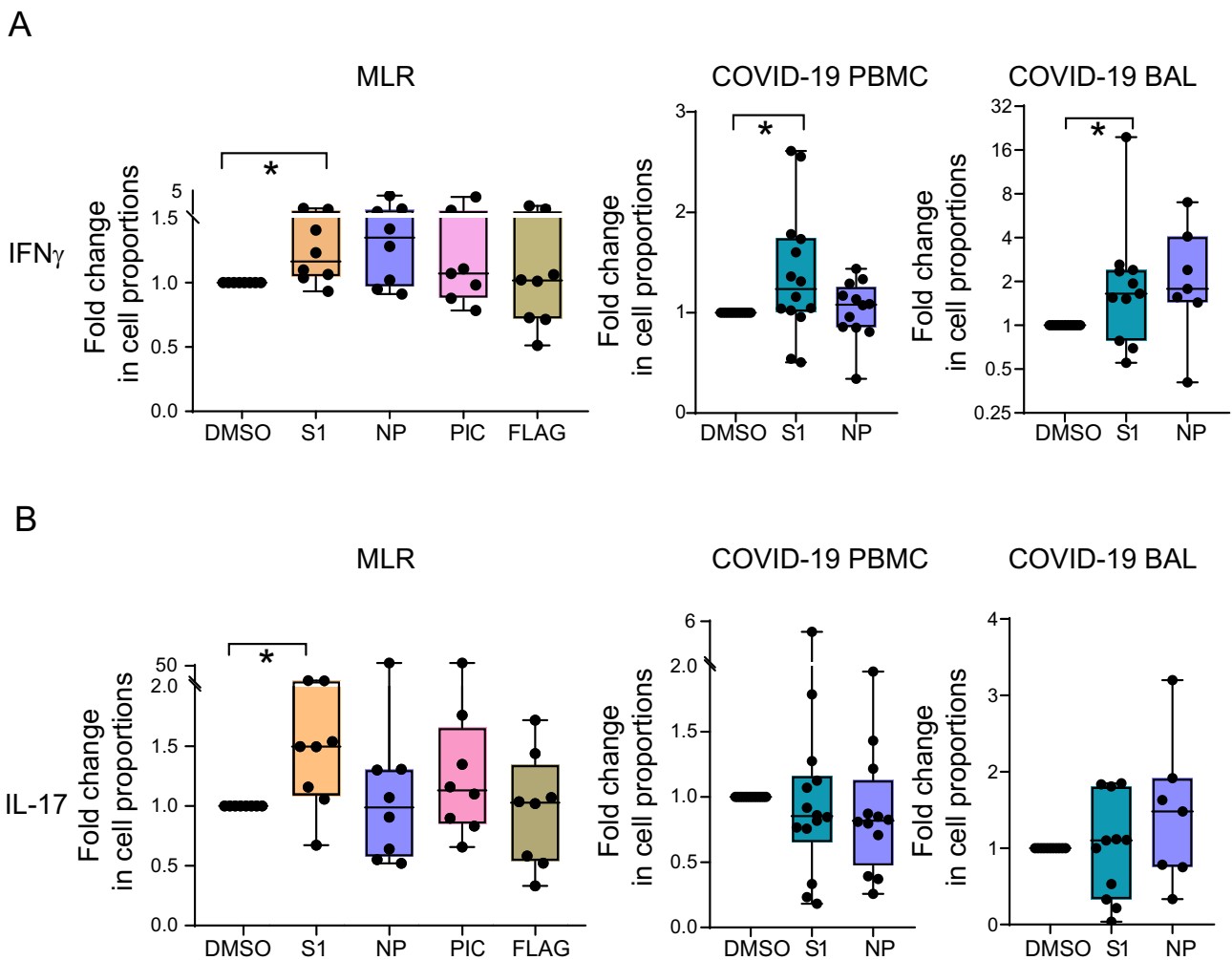

**Fig. 5 | Functional ability of Mo primed with SARS-CoV-2 proteins to activate IFNγ+ and Th17 CD4 + T cells and detection of specific responses in critical COVID-19 patients.** Fold change in proportions of CD4 + T cells expressing IFNγ (**A**) (*p* = 0.0391; 0.0367; 0.0137) and IL-17 (**B**) (*p* = 0.0391) either after 3 days of culture with allogeneic Mo (*n* = 8) (left panels) with DMSO or S1 (orange), NP (purple), Poly I:C (PIC; pink), Flagellin (FLAG, khaki) or induced directly by S1 (green) or NP (purple) peptide stimulation in PB (*n* = 14) (middle panel) and BAL (*n* = 11) (right panel) of severe COVID-19 patients. Data are represented as box and whiskers with bars representing maximum and minimum values and with median highlighted as a line. Statistical significance was calculated using a two-tailed Wilcoxon test: \**p* < 0.05.

these myeloid subsets, the induction of pathogenic T cells in the lung of COVID-19 patients or clinical parameters in these individuals. To this end, we first analyzed the presence of IFNγ and IL-17 producing cells in lung sections from critical COVID-19 patients. As shown in Fig. 6A, we observed that IL-17+ and IFNγ + T cells were found in highly infiltrated areas in the lung from critical COVID-19 patients consistent with our previous in vitro functional assays. We next correlated multiple clinical parameters as well as proportions of different Mo subsets at baseline and after stimulation with S1, and the presence of pathogenic CD4+ and CD8 + T cells in lungs from all tested COVID-19 patients (Fig. 6B, Supplementary Fig. 6A) and also in those BAL samples specifically inducing IL-17 responses upon S1 or NP stimulation (Fig. 6B, Supplementary Fig. 6B). Interestingly, higher basal proportions of NC-Mo infiltrated in BAL significantly correlated with higher levels of C Reactive Protein (CRP) and Procalcitonin (PCT) at admission as well as with fewer days of severe COVID-19 patients from symptoms onset to hospitalization and from hospitalization to transfer to ICU (Fig. 6B, C, Supplementary Fig. 6A). However, NC-Mo proportions were not higher specifically in BAL from patients that died at ICU (Supplementary Table 6). Moreover, higher basal frequencies of NC-Mo correlated with an even higher enrichment of these cells in response to S1 peptide in BAL samples (Fig. 6B, C, Supplementary Fig. 6A), suggesting a

preexisting trained state in this subset in the respiratory tract of patients with the most critical conditions. Interestingly, higher detection of Th17 CD4 + T cells at baseline significantly correlated with higher basal level of detection of IFNγ+ cells both in CD4+ and CD8 + T cells and also, with increased Th17 cells in response to S1 stimulation in BAL samples (Fig. 6B–D). Also, higher basal Th17 cell detection significantly correlated with higher levels of ferritin at admission, a clinical parameter associated with IL-1β hyperexpression[65] (Fig. 6B). Levels of fibrinogen at admission, which has also been linked with inflammasome during viral infections[66] tended to be associated with higher detection of PCT and CRP at this time point, and also correlated with higher proportions of NC-Mo after stimulation with S1 (Fig. 6B). On the other hand, levels of triglycerides at admission seemed to follow an opposite trend and inversely correlated with PCT, CRP and increase of NC-Mo after S1 exposure but were associated with higher levels of CD38 on T cells in the lung (Supplementary Fig. 6A, B). Consistent with our previous MLR results higher detection of NC-Mo positively correlated with higher IFNγ + CD4T cells after S1 stimulation, specifically in BAL samples from patients inducing Th17 responses (Fig. 6B). At the clinical level, when considering BAL from all critical COVID-19 patients we did not observe any direct significant correlation of basal or S1-induced Th17 cells with inflammation plasma reactants

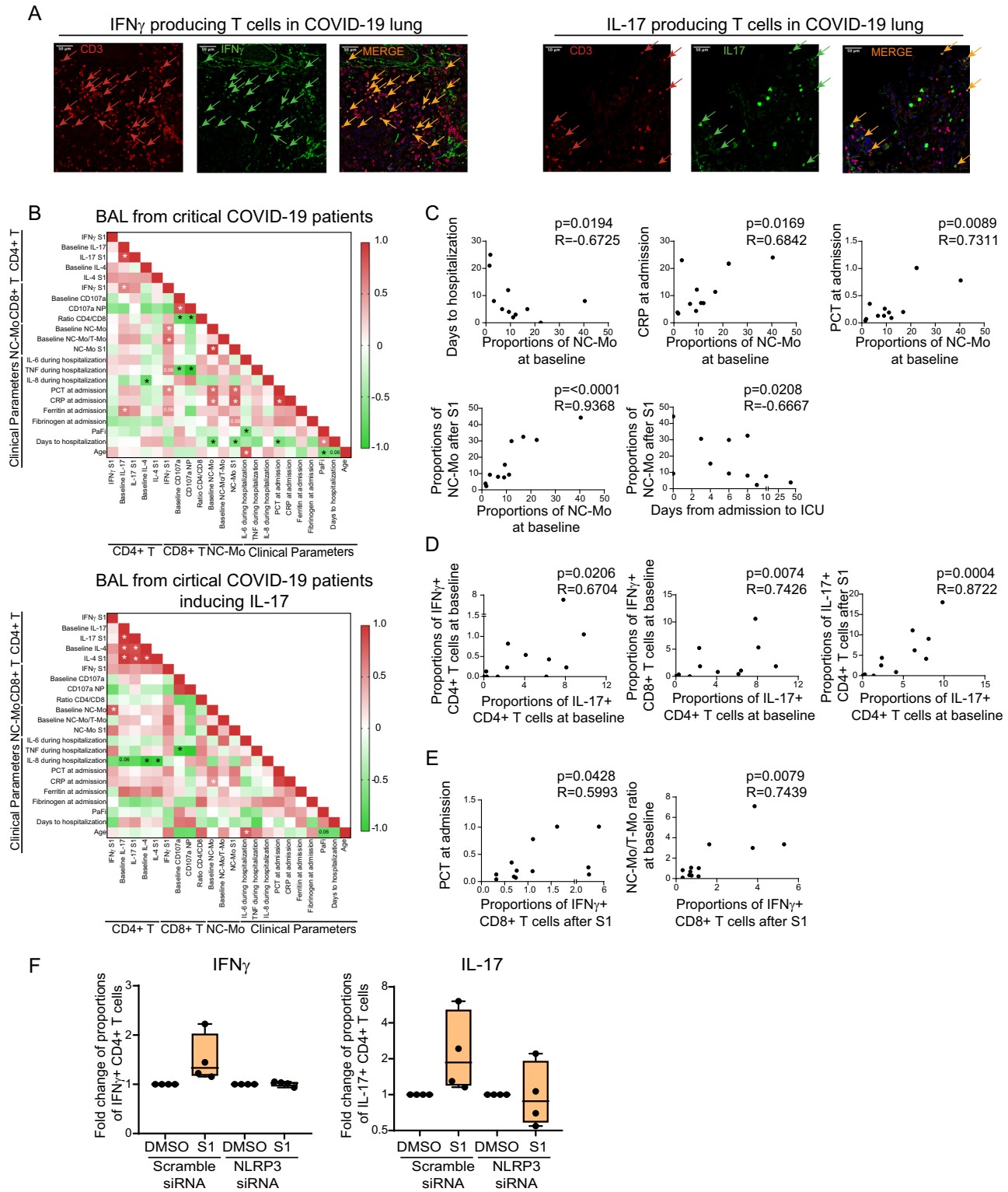

such as PCT, CRP, IL-6, IL-8, TNFα or time to hospitalization (Fig. 6B, Supplementary Fig. 6A, B). However, basal and S1- induced IL-17 expression significantly associated with the presence of fungal but not bacterial superinfection when considering all critical COVID-19 patients at ICU using a univariate analysis (Supplementary Table 6). Of note, fungal superinfection also tended to associate with mortality although not significantly (Supplementary Table 6). In addition, IL-4 detection at baseline and after S1 stimulation in BAL samples significantly correlated in patients with higher Th17 cells at baseline and

after S1 stimulation (Fig. 6B). Interestingly, significantly higher basal IL-4 detection in BAL was detected in BAL from patients that did not survive in our study (Supplementary Table 6) and was negatively correlated with levels of IL-8 in plasma (Fig. 6B). Therefore, our data support an association between NC Mo at baseline and the Th17 and Th2 cells in response to S1 with the severity of pathology critical COVID-19 patients.

Finally, we also asked whether there was an association between myeloid cell subsets and specific patterns induced by SARS-CoV-2 NP

**Fig. 6 | Association of Monocyte subsets with CD4+ and CD8 + T cell profiles in the lung of COVID-19 and with clinical parameters of severity. A** Representative confocal microscopy images showing analysis of IFNγ (left panels, green) or IL-17 (right panels, green) expression on CD3 + T (red) cells present in the lung tissue of three critical COVID-19 patients tested *n* = 3. **B** Heatmaps showing Spearman correlation networks of relevant parameters selected from networks in Supplementary Fig. 6 including all BAL samples from *n* = 12 COVID-19 patients tested (upper heatmap) or exclusively those *n* = 7 individuals in which IL-17 induction was observed in response to SARS-CoV-2 S1 or NP peptide stimulation (lower heatmap). Colors represent positive (red) or negative (green) R values of association. Associations reaching (**p* < 0.05) or close (*p* = 0.06) to statistical significance are highlighted in both heatmaps. **C, D** Individual Spearman correlations highlighting associations between basal NC-Mo proportions with frequency of the same

population after stimulation with S1 peptide (*p* < 0.0001) and with clinical severity parameters such as number of days to hospitalization (*p* = 0.0194), number of days from hospital admission to ICU (*p* = 0.0208) or plasma levels of procalcitonin (PCT) (*p* = 0.0089) and C Reactive Protein (CRP) at admission (*p* = 0.0169). Correlations between detection of Th17 cells at baseline in BAL and basal detection of IFNγ + CD4+ (*p* = 0.0206), CD8+ (*p* = 0.0074) T cells in BAL at baseline or S1-induced detection of IL-17+ cells (*p* = 0.0004) (**D**). **E** Associations between IFNγ + CD8 T cells after S1 stimulation with plasma PCT levels at admission and with basal NC-Mo/T-Mo ratios in tested COVID-19 BAL samples. **F** Fold change in proportions of IFNγ+ (left plot) or IL-17+ (right plot) CD4 + T cells after culture in the presence of allogeneic Mo nucleofected with scramble or NLRP3-specific siRNA for 3 days (*n* = 4). Data are represented as box and whiskers with bars representing maximum and minimum values and with median highlighted as a line.

and S1 in CD8 + T cells from BAL of severe COVID-19 patients. Interestingly, higher levels of IFNγ + CD8 + T cells in response to S1 associated with higher plasma levels of PCT at admission, higher concentration of TNFα in plasma and with higher relative proportions of NC-Mo vs T-Mo at baseline (Fig. 6B, E). No significant association between levels of CD107a induced by S1 or NP peptides were observed for most of parameters except for a negative correlation with CD4/CD8 ratios in BAL samples and plasma levels of TNFα during hospitalization (Fig. 6B, Supplementary Fig. 6A). Therefore, NC-Mo and clinical severity may correlate with IFNγ + CD8 + T cells in BAL from COVID-19 patients.

Finally, we focused on determining whether the induction of IFNγ + /IL17 + CD4 + T cells and IFNγ+/CD107a CD8 + T cells in response to Mo exposed to S1 and NP was dependent on activation of the inflammasome. To this end, we repeated the MLR in vitro functional assays using Mo treated with NLRP3 and NLRC4-specific siRNAs. As shown in Fig. 6F and Supplementary Fig. 6C, induction of IFNγ + CD4 + T cells by Mo exposed to S1 was abrogated after knock-down of NLRP3 expression. Similar findings were observed for Th17 cells (Fig. 6F, Supplementary Fig. 6C). In addition, expression of IFNγ and CD107a tended to be restored in CD8 + T cells exposed to Mo treated with NLRC4-specific siRNAs and stimulated with NP (Supplementary Fig. 6D). Together, these data suggest that inflammasome NLRP3/ NLRC4 sensors may participate in the functional specialization of Mo after S1 and NP exposure.

## Discussion

Our study provides missing links between molecular mechanisms involved in the innate sensing of different SARS-CoV-2 proteins by monocytes, the differentiation and activation patterns in different Mo and how these responses associate with adaptive T cell profiles present in COVID-19 patients. Our data indicate that SARS-CoV-2 S1 and NP proteins can distinctly regulate Mo differentiation and activation through independent inflammasome and NFκB pathways. Hence, S1 efficiently enhances the differentiation from the C-Mo subset into T-Mo and then to NC-Mo subset, whose frequencies are altered in the peripheral blood (PB) and in the lung from infected patients with different degree of pathology[21]. These data are particularly relevant since Mo are central players regulating innate and adaptive immunity[67,68]. Recruitment and mobilization of Mo play an essential role in anti-viral and anti-bacterial immune responses, and their migration to tissues is required for the correct priming of CD4+ and CD8 + T cells and the control of viral and bacterial replication in these sites[69,70] where they can also differentiate into macrophages and dendritic cells[14,19,20].

Mo subsets are believed to be differentiated in sequence from C-Mo to T-Mo and eventually, NC-Mo[71,72] directly in the bloodstream[72] or after migration to tissues, and once differentiated they can return to circulation[73]. In our study, comparing the efficiency of Mo differentiation into T and NC subsets between cell cultures in bulk and in Mo-enriched cultures, the differentiation in response to S1 is much more efficient when Mo are cultured in the presence of other cells. This

might suggest that the differentiation from T-Mo into NC-Mo might require additional signals provided by neighboring cells. However, we cannot rule out that in bulk the preexisting NC-Mo could survive at a higher rate in the presence of S1 and die in the absence of other stimuli. To address this issue, further research either performed with sorted T-Mo in the presence or absence of cell-to-cell contact or with Mo previously depleted of NC-Mo subsets should be conducted. Remarkably, we observed a consistent increase in T-Mo proportions and a reduction the C-Mo subset upon the S1 stimulation of bulk cultures and isolated Mo, supporting a role in the differentiation. This differentiation process was mediated by the NFκB signaling pathway according to the result obtained in the presence of parthenolide. In contrast, other inhibitors directed against inflammasome proteins did not appear to exert any significant effect on the Mo differentiation, thereby indicating that the differentiation process may be inflammasome-independent. We validated these data in BAL samples from lung infiltrates of COVID-19 patients. Therefore, NFκB pathway may most likely regulate NC-Mo generation in the lung during SARS-CoV-2 infection. Accordingly, it was reported that NFκB-induced genes are upregulated upon S-protein stimulation in macrophages and NFκB inhibitors are able to suppress IL-1β secretion induced by S1 protein[25]. Moreover, it was previously reported that the transcription factor NFκB was enriched in T Mo subset[18]. The NFκB pathway is responsible for a vast variety of pathways including the inflammatory response induced by TLR signaling. In this line, our results indicate that in addition to IL-1β expression, SARS-CoV-2 S1 also induced the transcription of other NFκB-induced cytokines such as IL-8, in agreement with independent studies involving TLR2 in S protein of SARS-CoV recognition[74]. Interestingly, we observed different effects of viral proteins in transcription of other cytokines mediating inflammasome activity such as IL-18, which was not induced in response to viral proteins, in contrast to IL-1β. Such results are compatible with previous studies suggesting that IL-18 transcription is dependent on IFN[55] and may be subject of different regulatory mechanisms compared to IL-1β[75]. These findings are potentially relevant for clinical applications, as NFκB inhibitors could be used as a therapeutic tool to limit Mo differentiation as well as controlling inflammatory cytokine storm during COVID-19, as recently proposed by other groups[76]. In fact, parthenolide has already been tested as an antitumor therapy to reduce NFκB activation in cancer[77]. A limitation of our study is that we did not directly address which TLR might be involved in the process or studied in more detail the activation of NFκB in response to SARS-CoV-2 S1. We observed that TLR5 but not TLR3 stimulation induced a similar Mo differentiation pattern to that induced by S1. Previous studies reported that S-protein can be recognized and induce innate activation via TLR2[28], while others claim that TLR4 rather than TLR2 is accounting for such response in cooperation with CD14 and MD2[25]. TLR4 is the canonical receptor of LPS and signals through NF-κB pathway; it has been described to regulate Mo differentiation into macrophage and Mo subsets in the lung from mice infected with *Streptococcus pneumoniae*[78]. Therefore, our observations bring up the possibility

that alternative TLRs or additional molecules involving NFκB are responsible for Mo differentiation in response to SARS-CoV-2 S1. Further studies are needed to delineate which TLR are mediating the S1 effect using specific inhibitors or siRNAs directed to individual receptors. On the other hand, the three Mo subsets differ in their intrinsic basal expression of CD40 and CD86[16], and the decrease of CD40 expression observed in T-Mo exposed to SARS-CoV-2 could reflect either the transition of activated CD40+ cells into the next NC-Mo differentiation stage or be connected with the inability of T-Mo to increase CD40 beyond their basal maximal levels. However, the increase in costimulatory molecule levels in NC-Mo and T-Mo subsets after the stimulation with SARS-CoV-2 S1 and NP proteins could also be related to the induction of alternative pathways affecting specifically these molecules.

Regarding the involvement of the inflammasome in the response of Mo to SARS-CoV-2, our data suggest that differential effects observed after siRNA-mediated silencing of NLRP3 and NLRC4 may support the view that these sensors play different roles in the innate detection of different viral proteins and the induction of Mo activation. We cannot completely rule out that the induction of the inflammasome and the secretion of IL-1β after exposure to S1 are independent from the differentiation of Mo into T-Mo, as it was reported that LPS-stimulated T-Mo and NC-Mo can produce larger amounts of TNF-α and IL-1β[16,28]. Moreover, detection of TNF-α, IL-8 and clinical parameters associated to inflammasome activation are increased in plasma from severe COVID-19 patients and correlate with the induction of NC-Mo in the lung in response to S1. While our findings are consistent with an association of the inflammasome with COVID-19 severity as previously suggested[79], a recent clinical trial and metanalysis in large cohorts did not find a consistent benefit on mortality after of the use of IL1R antagonist (Anakinra) in COVID-19 expressing increased levels of the biomarker soluble urokinase plasminogen activator receptor (suPAR)[80–82], a parameter which is associated with mortality and comorbidities[83]. However, some studies may support some benefit of anakinra reducing the need for invasive mechanical ventilation at ICU[84] and contributing to reduce levels off inflammatory plasma biomarkers[85] and inflammasome activation in Mo[86] from critical COVID-19 patients. In fact, the use of anakinra has been approved for COVID-19 patients in Europe and US[87,88]. These findings suggest that the contribution of the inflammasome may be determinant at earlier phases of the pathology, before hospitalization or may require specific targeting of cell populations in which this pathway is activated. In this regard, our study provides important knowledge about key cell populations, in which the inflammasome and NFκB pathways are induced in response to different viral proteins. Therefore, the activation of these pathways could be selectively targeted in specific Mo populations, which therefore could lead to the development of future directed therapies against SARS-CoV-2.

In addition, we have observed increased detection of cleaved Gasdermin D in Mo exposed to S1, which could be involved in the secretion of IL-1β after activation of the NLRP3 inflammasome. While these results are compatible with the canonical and non-canonical NLRP3 activation pathways, more studies should focus on the precise mechanisms leading to IL-1 β secretion and the involvement of Gasdermin-D. In fact, NLRP3 inflammasome can induce IL-1 β secretion independently of Gasdermin-D[89]. On the other hand, we observed that the NP involves the inflammasome but might be more dependent on NLRC4 than on NLRP3 to regulate CD86 expression in C-Mo and T-Mo, which contrasts to previous studies highlighting the role of NLRP3[90]. This discrepancy might be related to the fact that the authors determined the participation of NLRP3 based on the oligomerization of ASC in a fibroblast cell line transfected with different NLRP3 and NLRC4 but did not validate the activity of these inflammasome in response to NP on primary innate myeloid cells such as monocytes. Nevertheless, it is remarkable that virus-derived proteins are able to trigger a sensor such

as NLRC4, which was initially described as molecule participating in detection of bacterial proteins such as flagellin and T3SS[34]. In addition, induction of NLRP3 inflammasome could be triggered after the TLR activation, as it was previously reported[91,92], suggesting that NLR sensors might have a broader repertoire of sensing than initially anticipated[25]. Interestingly, induction of inflammasome and IL-1β secretion by Mo might be dependent on previous sensibilization to SARS-CoV-2[28], however, we did not address this possibility in our study. While it has been established that TLR-dependent inflammasome activation requires a second activation signal such as ATP release[93], our results suggest that stimulation of Mo with S1 and NP leads to complete cleavage of procaspase 1. In addition, we have shown that the NLRP3 sensor co-distributes with caspase-1 after S1 stimulation, suggesting the formation of the inflammasome complex in Mo. In contrast, while we were not able to observe co-distribution of NLRC4 with caspase-1, we did observe that siRNA-mediated knock down of this sensor reduces activation of Mo in response to NP. These discrepancies may be due to different kinetics of association of this sensor with caspase-1 or other caspases, such as caspase-4[94]. Therefore, more detailed molecular biology studies describing direct interactions between different caspases and inflammasome sensors are needed. In addition, alternative inflammasome sensors involved in inflammation may also participate in the pathogenesis of COVID-19[95,96]. Another limitation of our siRNA silencing assays is the fact that we used mRNA levels as a readout for efficacy of NLR gene knockdown. However, since this may exclusively affect the newly synthesized proteins from mRNA and the turnover of these molecules might be different, downregulation of the protein sensor levels should be confirmed as well. In addition, since the efficiency of NC-Mo generation from isolated Mo is lower than in bulk cultures, we did not obtain conclusive results regarding the role of NLR in the activation of this particular subset. Therefore, future studies using isolated NC-Mo should be conducted. An important point to discuss is the fact that in our study we chose to study the impact of S1 and NP on Mo differentiation and activation individually, without considering whole viral particles, or other viral proteins such as nsp1, ORF3b, ORF6 and ORF8, which have been shown to modulate innate immune responses such as IFNs against SARS-CoV-2 or other coronaviruses[97–99]. Despite these limitations, our study extends current knowledge on the role of different SARS-CoV-2 proteins inducing different inflammasomes and NFκB pathways in primary Mo from peripheral blood and BAL samples, providing additional information that may be relevant for clinical management, the modification of current therapies to severe COVID-19 or potential side effect after immunization with SARS-Co-V2 S1.

Finally, we showed that priming of Mo by either S1 or NP may differentially affect their functional ability to modulate CD4 + T versus CD8 + T cells. In this regard, this study provides a functional link between innate activation of Mo in response to SARS-CoV-2 proteins and their ability to activate T cell responses involved in the pathogenesis of severe COVID-19. Mo can act as antigen presenting cells (APC) and T-Mo express high levels of Human Leukocyte Antigen-class II (HLA-DR), which allows them to mediate efficient activation and polarization of CD4 + T cells in vitro[16,17] and during viral infections[37]. We have shown that S1-primed Mo promote high levels of IFN-γ expression on CD4 + T cells. IFN-γ producing CD4 + Th1 cells are typically associated to viral control[100], but they can also participate in pathogenic inflammation of airways and in the context of autoimmunity[46,101]. In fact, we have observed that Mo exposed to S1 induce pathogenic-like cells co-expressing high levels of IFN-γ and high levels of IL-4 at least in vitro. However, we found IFN-γ secretion induction by CD4 + T cells upon exposure to SARS-CoV-2 S1 stimulation, confirming that T-Mo and NC-Mo subsets may play a role[46]. We included IL-4 in our staining panel due to the association of IL-4 induction with the progression of idiopathic pulmonary fibrosis[102].

IL-4 is a cytokine enhancing the Th2 immunity and inhibits Th1 immunity, also inhibiting IFN-γ production. However, there are studies reporting the Th2 response in COVID-19 patients[103]. While IL-4 expression is detected at high levels in our COVID-19 BAL samples and also in CD4 + T cells from the lung of COVID-19 patients stimulated with SARS-CoV-2 peptides[46], we did not observe any consistent induction of cells expressing IL-4 individually or in combination with IFN-γ after exposure to SARS-CoV-2 proteins in our BAL and PBMC samples from severe COVID-19 patients. A potential reason for these discrepancies may be related to differences in the assays considering that priming in MLR assays taking several days of culture, which may un-specifically affect the T lymphocytes. In contrast, peptide stimulation of bulk PB and BAL cultures from COVID-19 patients was induced by antigen presentation by autologous cells for only 7 h of incubation, while other groups have performed overnight peptide stimulations[43,104]. Therefore, further optimization of the kinetics and resolution of intracellular detection of these cytokines should be conducted in BAL samples upon antigenic stimulation. In addition, IL-4 plays a role by limiting the influx of pathogenic T cells and it is tempting to speculate that this mechanism could have failed in those patients[102]. Supporting this possibility, we have observed an association between basal detection of IL-4 in BAL samples and PCT levels in critical COVID-19 at ICU. Moreover, Th2 polarized response is associated with the pulmonary fibrosis development[105]. It also was reported that Th2 pathogenic cells participate in pathogenesis of bronchial asthma and chronic dermatitis by expression of IL-5 that recruits eosinophiles and IFN-γ producing cells in viral infections[105–107]. Therefore, the role of IL-4 in pathogenic CD4 + T cell responses during SARS-CoV-2 infection needs to be studied in more detail. Importantly, we have observed that proportions of CD4 + T cells secreting other inflammatory cytokines such as IL-17, were induced upon the stimulation with S1-primed Mo in MLR assays which is highly expressed at baseline in BAL from COVID-19 patients and induced at least in half of these samples after S1 peptide stimulation, suggesting the role of primed Mo in the induction of such pathogenic T cells. Of note, siRNA mediated-knock down of NLRP3 abrogated the activation of IFNγ+ an and IL-17 + CD4 + T cells in MLR assays, providing a mechanistic link for the functional specialization of Mo in response to S1. These data are in accordance with previous works reporting a role of Th17 cells in lung injury and an upregulation of this subset in ARDS, suggesting it as a biomarker of severe COVID-19[41,108,109]. Moreover, our results also suggest that higher enrichment on NC-Mo in BAL samples associates with induction of Th17 responses, which also matches our previous observations in MLR assays, suggesting a pathogenic role of NC-Mo during critical COVID-19. In this regard, although NC-Mo were initially considered suppressive with tissue healing functions[110,111], they also display high levels of expression of Tumor Necrosis Factor α (TNF-α) and IL-1β in response to Lipopolysaccharide (LPS) stimulation[17,112]. NC-Mo have been previously linked to high TNF expression and immune hyperactivation during Human Immunodeficiency Virus (HIV) infection[113]. However, the induction of IL-17 responses in cells from BAL samples may be in part due to activation of gamma-delta T cells, which are not present in our MLR assays but have been reported to play a role in severe COVID-19[64,114]. In addition, our data is not incompatible with the potential implication of additional immune cell types present in the blood and in the lung during the pathogenesis of severe COVID-19 such as neutrophils[115,116], DCs[117] and B cells[118]. In this regard, future studies should address the connections between innate sensing of SARS-CoV-2 by Mo and the pathogenic responses potentially induced by these additional cell types.

Finally, our study suggests that the impact exerted by Mo on CD8 + T cells is more limited and might be restricted to modulation of cytotoxic T cells after priming in response to NP. We observed a reduction of CD107a in CD8+ cells in response to NP, implying the reduction of the cytotoxic activity. The same effect was also observed in PB and BAL from critical COVID-19 patients, which suggest that this phenomenon might be also related to the immune exhaustion of those cells[119]. This interpretation is in line with lower expression of CD107a on CD8 + T cells expressing high levels of CD38 enriched in COVID-19 lung[21]. Interestingly, CD38 is a marker associated with hyperactivation, immune exhaustion and higher expression of checkpoint receptors in COVID-19[120] and in HIV-1[121] infections. In addition, our data of reduction of CD107a expression of CD8 + T cells from PB and BAL from severe COVID-19 patients is consistent with previous data from other studies that report decreased CD107a expression in CD8+ from these patients after stimulation with SARS-CoV-2 NP peptides[46]. Importantly, we detected that the decrease of IFN-γ- CD107a+ cells is marked both in CD103+ and CD103- CD8 + T cells, which seems to validate previous reports highlighting the importance of tissue resident cells in the antiviral response. However, CD103- CD8 + T cells from BAL infiltrates may also participate in the pathology and should be further explored[48]. In contrast, we did not observe a consistent effect of NP on IFN-γ expression after stimulation with NP. In contrast, S1 did increase IFN-γ production in CD8 + T cells from COVID-19 PBMC and BAL which seemed to correlate with Th17 responses induced by this viral protein. Thus, enhanced regulatory cytokine secreting function CD8 + T cells may be differentially promoted by S1 and NP proteins and may affect the induction pathogenic CD4 + T cells. Future studies should elucidate the connections between the two adaptive immune responses.

Together, this study shows that SARS-CoV-2 proteins such as S1 and NP differentially contribute to the Mo differentiation and activation patterns. The differentiation driven by S1 requires activation of the NFκB pathway, while the activation of C-Mo and T-Mo in response to S1 and NP is mainly dependent on NLRP3 and NLRC4 inflammasome signaling, respectively. Finally, these differentiation and activation patterns affect the Mo functionality by differentially promoting their ability to induce the activation of different subsets of CD4+ *versus* CD8 + T lymphocytes that might be relevant for COVID-19 pathology.

## Methods
### Study cohorts
$n = 17$ samples of peripheral blood mononuclear cells (PBMC) and $n = 17$ bronchoalveolar lavage (BAL) from severe and critical COVID-19 patients, a total of $n = 100$ PBMC samples from healthy donors ($n = 70$ Buffy Coats provided by Centro de Transfusiones de Comunidad de Madrid, and $n = 30$ volunteers from our hospital) and $n = 6$ non-COVID-19 control BAL samples from cancer (50%) and interstitial lung disease (50%) patients were recruited for the study using informed consent (reference 4381) approved by the bioethical committee from Hospital Universitario La Princesa de Madrid following principles of the Helsinki declaration. COVID-19 patient and control donor samples were used in different sets of experiments. Clinical parameters including plasma inflammatory reactants, parameter associated with hypoxia such as $PaO_2/FiO_2$ (PaFiO2), microbial superinfection, mortality, and demographic data of recruited COVID-19 patients were collected in a database created by the Immunology and the Pneumology Units from Hospital de la Princesa (Supplementary Tables 1–3).

### Immunomagnetic isolation of Mo
Circulating CD14+ Mo were purified from PBMC of healthy donors by incubation with CD14 Microbeads (Myltenyi Biotec) for 30 min in Running Buffer (autoMACS® Running Buffer – MACS® Separation Buffer, Miltenyi Biotec) and positively magnetic selection using MS Columns (Miltenyi Biotec).

### Stimulation of PBMC or isolated Mo with SARS-CoV-2 proteins
Preisolated CD14+ Mo or Mo present in bulk PBMC or BAL from healthy donors and COVID-19 patients, were cultured in RPMI medium (Gibco™ RPMI 1640 Medium, GlutaMAX™ Supplement, Fisher Scientific) supplemented with penicillin and streptomycin and with 10% of bovine fetal

serum (Fetal Bovine Serum, Collected in South America, HyClone™) with either 0.75 µg/ml SARS-CoV-2 Spike-1 peptide (S1, PepMix™ SARS-CoV-2 (Spike Glycoprotein), JPT Peptide Technologies) or 1 µg/ml Nucleoprotein (NP, PepMix™ SARS-CoV-2 (NCAP), JPT Peptide Technologies) pools of peptides or with the equivalent volume of filtrated dimetilsulfoxid (DMSO, Panreac) for 16 h. For comparison purposes, Mo were also cultured in the presence of ligands for different TLRs including 1.25 µg/ml TLR5 (Flagellin, Invivogen), 2.5 µg/ml TLR3 (Poly I:C, MERCK). In some experiments, isolated Mo were also cultured in the presence of 1 µM Parthenolide (NF-κB and Caspase-1 inhibitor, Invivogen), 2 µM MCC950 (NLRP3 inhibitor, Invivogen), 2 µg/mL Z-VAD-FMK (Pan-caspase inhibitor, Santa Cruz Biotechnology).

## SiRNA mediated knock down of inflammasome sensors

Primary Mo preisolated from PBMC of healthy donors were nucleofected with siRNA specific for NLRP3 and NLRC4 and non-targeting control pool (Dharmacon) using a Nucleofector™ 4D and P3 Primary Cell kit (Lonza) using the manufacturer's protocol. Nucleofected Mo were cultured overnight with medium containing 10% FBS without antibiotics and gene knockdown was confirmed by RT-qPCR.

## RT-qPCR quantification of transcriptional expression of proinflammatory cytokine and inflammasome sensors

Mo were lysed with RLT buffer (RNA KIT RNeasy Micro Kit Qiagen), and total RNA was extracted using the RNA KIT RNeasy Micro Kit (Qiagen). cDNA was synthetized by using Promega kit in total volume of 25 µl. The resulting cDNA was then amplified by qPCR with primers specific for *ACTB* (β-actin) Fw: 5′ CTGGAACGGTGAAGGTGACA 3′ Rv: 5′ CGGCCACATTGTGAACTT 3′, *IL1B* (IL-1β) Fw: 5′ ATGATGGCTTATTCAGTGGCAA 3′ Rv: 5′ GTCGGAGATTCGTAGCTGGA 3′, *IL6* Fw: 5′ CCTGAACCTTCCAAAGATGGC 3′ Rv: 5′ TTCACCAGGCAAGTCTCCTCA 3′, *CXCL8* (IL-8) Fw: 5′ TCT GTG TGA AGG TGC AGT TTT G 3′ Rv: 5′ GGG GTG GAA AGG TTT GGA GT 3′, *IL18* Fw: 5′ TCTTCATTGACCAAGGAAATCGG 3′ Rv: 5′TCCGGGGTGCATTATCTCTAC 3′, *TNF* Fw: 5′ CAGCCTCTTCTCCTTCCTGAT 3′ Rv: 5′ GCCAGAGGGCTGATTAGAGA 3′, *IFNB1* (IFNβ) Fw: 5′ GAATGGGAGGCTTGAATACTGCCT 3′ Rv: 5′ TAGCAAAGATGTTCTGGAGCATCTC 3′, *NLRP3* Fw: 5′ GATCTTC GCTGCGATCAACA 3′ Rv: 5′ GGGATTCGAAACACGTGCATTA 3′ and *NLRC4* Fw: 5′ GGCAATTGGATTGCTCAGCC 3′ Rv: 5′ GGAAAGGTCAAAGGTGATCCCA 3′ using GoTaq® qPCR Kit (Promega) and data was obtained and analyzed by StepOne™ Software V2.3 (Applied Biosystems).

## Mo protein quantification by Western Blot assays

Isolated Mo cultured at 500,000 cells/well in the presence of different stimuli were lysed using RIPA buffer 1% of phosphatase and protease inhibitors (Roche Diagnostics) with 10 min of sonication. Protein lysates were resolved in a 10% acrylamide (30% Acrylamide/Bis Solution 29:1, Bio Rad) gel with SDS and transferred to a nitrocellulose membrane (Fisher Scientific). Afterwards, membranes were blocked with 4% BSA (Sigma-Aldrich) in Tris-Buffered saline and incubated at 4 °C overnight with primary antibodies directed either to phospho-p65 (Cell Signaling ref 3039), anti-p65 (ab32536, E379, Abcam;), anti-caspase-1 (AF6215; R&D Systems), anti-Gasdermin D (69469, E5O4N; Cell Signaling), anti-IL-1β (12242, 3A6; Cell Signaling) or anti-GAPDH (FF26A/F9; BioLegend) following manufacturer's specifications. Each primary antibody was used at a 1:1000 dilution. Subsequently, the membranes were washed and incubated at RT for 1 h with the respective anti-rabbit (31460; Thermofisher; dilution 1:2000), anti-goat (81–1620; Thermofisher; dilution 1:2000) or anti-mouse (31430; Thermofisher; dilution 1:5000) secondary antibodies. Chemiluminescence of protein band intensity was quantified by ImageQuant 800 system (Amersham) using the IQ800 V1.2.0 Software.

## Cytokine quantification

IL-1β concentration in culture supernatants of Mo exposed to different SARS-CoV-2 proteins was evaluated using the Human IL-1 beta/IL-1F2 Quantikine ELISA Kit (Biotechne, R&D Systems) and analyzed in a GloMax® Discover Microplate Reader (Promega). To determine to concentration of cytokines as IL-8 and TNFα from plasma of COVID-19 patients, Human Inflammatory Cytokine CBA (BD™ Cytometric Bead Array) was used following the manufacturer's instructions.

## Mixed Lymphocyte Reaction assays

Isolated Mo ex vivo or nucleofected with scramble or NLRP3 or NLRC4-specific siRNAs and previously cultured with different SARS-CoV-2 proteins or DMSO were co-cultured with allogenic total T lymphocytes isolated from the PBMC from healthy donors by negative selection using the Dynabeads™ Untouched™ Human T Cells Kit (Invitrogen) at a 1:5 ratio in the presence of RPMI medium with antibiotics and 10% FBS supplemented with 100 IU/ml recombinant human IL-2 (Stem Cell Technologies). Fresh medium and cytokines were added every 2 days and phenotype of T cells was analyzed at day 3 of culture by flow cytometry.

## Flow cytometry analysis of Mo and co-cultured CD4+ and CD8 + T Lymphocytes

For the phenotypical characterization of Mo isolated from PBMC and BAL, cells were blocked with 100 µg/ml of IgG antibody (Sigma) and subsequently stained with a panel of fluorochrome-marked mAbs including anti-CD14-PE (BioLegend; 1:50 dilution), anti-CD40-FITC (BioLegend; 1:50 dilution), anti-CD86-PECy7 (BioLegend; 1:50 dilution), anti-CD16-Pacific Blue (BioLegend; 1:50 dilution) and APC-Cy7 Ghost red viability Dye 780 (Tombo Biosciences; 1:100 dilution) for 15 min at 4 °C. Samples were fixed and preserved in 2% paraformaldehyde for flow cytometry analysis. In some experiments, intracellular expression of IL-1β (anti-IL-1β-FITC BioLegend; 1:100 dilution) was assessed in Mo exposed to SARS-CoV-2 peptides, TLR ligands or DMSO after 16 h of culture in the presence of 2.5 µg/ml Brefeldin A (MERCK) and 2.5 µg/ml Monensin (Sigma).

For analysis of T cells polarization from MLR assays, lymphocytes were incubated for 1 h in presence of 0.05 µg/ml of PMA (MERCK) and 0.25 µg/ml of Ionomycin (Peprotech), and then for 4 h in presence of 5 µg/ml Brefeldin A (MERCK), 5 µg/ml Monensin (Sigma) and anti-CD107a-APC (BioLegend; 1:300 dilution). Cells were subsequently stained in different panels including Ghost Red viability Dye 780, anti-CD3 (Immunostep; 1:50 dilution), anti-CD8 (BioLegend; 1:50 dilution), anti-CD103 (BioLegend; 1:50 dilution), anti-CD3 (Immunostep; 1:50 dilution), anti-IFN-γ (BD Biosciences at 1:100 dilution or BioLegend at 1:50 dilution), anti-IL-4 (BD Biosciences; APC at a 1:100 dilution and PE at 1:30 dilution) and anti-IL-17 (BioLegend or Invitrogen, both at a 1:100 dilution) mAbs (Supplementary Table 4). Samples were acquired using a BD FACSCanto™ II (BD FACSDiva Software V9.0) and FACSLyric™ (BD FACSuite Software V1.5.0.925) cytometers (BD Biosciences). Data was analyzed by FlowJo™ Software (version 10.7.).

**Flow cytometry analysis of SARS-CoV-2-specific T cell responses in COVID-19 patients.** Cryopreserved PBMC and BAL samples from severe COVID-19 patients were stained at 300,000–500,000 cells/well plated in a 96 Well Flat Bottom plaque. Cells were rested in RPMI culture media and 10% FBS at 37 °C for 1 h and subsequently exposed to 0,5 µg/well of a mix of overlapping SARS-CoV-2 S1 or NP peptides or the same volume of DMSO as a negative control in the presence of 3 µg/mL of BD FastImmune aCD28/aCD49d mAbs (BD-Biosciences). Additional cell samples stimulated in parallel with

50 ng/ml PMA and 0.25 µg/ml Ionomycin were included as a positive control of cytokine induction. Those cells were incubated in the presence of the peptides for 2 h and subsequently, 5 µg/ml Brefeldin A, 5 µg/ml Monensin and 0.2 µg of anti-CD107a-BV510 (1:200 dilution) (Supplementary Table 4) were added per well in the culture. After 4 h, cells were stained and analyzed by flow cytometry using the intracellular staining panel for cultured T cells described in the section above.

## Immunofluorescence in Mo

$10^5$ isolated Mo cultured for 16 h in DMSO or S1 or NP SARS-CoV-2 proteins were placed in Poly L Lysine covered glasses and expression inflammasome components was carried out by immunofluorescence. Briefly, adhered Mo were fixated with 4% PFA, blocked with a PHEM solution containing 50µg/mL human immunoglobulins and 3% Bovine Serum Albumin (BSA). Then, cells were stained with rabbit anti-human NLRC4 (abcam; 1:500 dilution) or rabbit anti-human NLRP3 (Cell Signaling; 1:50 dilution) and with goat anti-human Caspase-1 (Bio-Techne RyD Systems; 1:200 dilution) as primary antibodies for 16 h at 4 °C. Subsequently, cells were incubated with donkey anti-goat AlexaFluor™ 568 (Invitrogen; 1:200 dilution), donkey anti-rabbit AlexaFluor™ 488 (Invitrogen; 1:400 dilution) secondary antibodies and DAPI (1:400 dilution) for 1 h at 37 °C (Supplementary Table 5). Images were obtained with a Leica TCS SP5 confocal microscope and processed with the LAS AF software. Images were analyzed using ImageJ software. The areas of co-distribution of NLRP3/NLRC4 sensors with caspase-1 was calculated combining values from each of 43 planes and 4-6 microscopy fields. Numbers of Mo on each field were identified by DAPI staining and used to normalize the data obtained for each field.

## Histological analysis of COVID-19 lung tissue

Lung biopsies from deceased critical COVID-19 patients embedded in paraffin and segmented in 3 µm fragments were provided by Dr. Palacios-Calvo. Tissue sections deparaffinization, hydration and antigen retrieval were performed with a PT-LINK (Dako) prior to Ab staining. Tissue slices were then stained with either a rabbit anti-human NLRC4 (abcam; 1:400 dilution), rabbit anti-human NLRP3 (Cell Signaling; 1:50 dilution) goat anti-human Caspase-1 (Bio-Techne RyD Systems; 1:100 dilution), mouse anti-human CD14 (abcam; 1:200 dilution), rabbit anti-human IFNγ (abcam; 1:100 dilution), goat anti-human IL-17a (Bio-Techne RyD Systems; 1:50 dilution) and mouse anti-human CD3 (Dako; 1:50 dilution) as primary antibodies; and donkey anti-rabbit AlexaFluor™ 488 (Invitrogen; 1:200 dilution), donkey anti-goat AlexaFluor™ 488 (Invitrogen; 1:200 dilution), and donkey anti-mouse AlexaFluor™ 647 (Invitrogen; 1:200 dilution) were used as secondary antibodies (See Supplementary Table 5). Images were obtained with a Leica TCS SP5 confocal microscope and processed with the LAS AF software. Co-localizations, visualization and quantifications of NLRP3 or NLRC4 inflammasomes with CD14 and caspase 1 or IL-17a or IFNγ and CD3 staining in histological images were analyzed in with ImageJ software.

## Statistical analysis

Statistical significance effects of different experimental conditions in the same experiment or the same donors compared to a control/basal condition, were analyzed using a non-parametric two tailed Wilcoxon pairs-matched test. When comparing samples from different cohorts we used a two tailed Mann–Whitney test. In some cases, Kruskal–Wallis followed by Dunn's post-hoc tests were used for multiple comparison analyses. Data from experiments in which technical or low viability issues occurred were excluded from the analysis. All statistical analyses were performed using the GraphPad Prism Software (version 8.0.0, www.graphpad.com).

## Reporting summary

Further information on research design is available in the Nature Portfolio Reporting Summary linked to this article.

## Data availability

The experimental data and the results that support the findings of this study are available in this paper and the Supplementary Figs. and source data are provided. Source data are provided with this paper.

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

## Acknowledgements

I.T. was supported by Fomento de Investigación and FPI-UAM fellowships by Universidad Autónoma de Madrid. E.M.G. and I.T. were funded by PID2021-127899OB-I00 Generación de Conocimiento and CNS2023-144841 consolidación investigadora grants from Agencia Estatal de Investigación. E.M.G. and C.D.A. were also supported by RYC2018-024374-I. Ramón y Cajal Program. E.M.G., M.J.B., and M.G. were supported by REDINCOV by la MARATÓ TV3 (202104-30-31) and La Caixa Foundation HR20-00218. E.M.G., I.S. and I.S.C. were funded by CIBERINFECC from Instituto de Salud Carlos III. O.P. was supported by REDINCOV. MCM was supported by La Caixa Banking Foundation LCF/PR/HR20-00218. IGA was supported by PEARL thanks to grants RD16/0011/0012 and PI18/0371 from the Ministerio de Economía y Competitividad (Instituto de Salud Carlos III) and co-funded by Fondo Europeo de Desarrollo Regional (FEDER). I.S.C. was supported by the Rio Hortega Grant program (CM21/00157) and CIBERINFECC from Instituto de Salud Carlos III. F.S.M., A.A. were supported by INMUNOVACTER REACT-EU grant from Comunidad de Madrid. F.S.M. was also supported by grants PDC2021-121719-I00 and PID-2020-120412RB-I00 from the Spanish Ministry of Economy and Competitiveness (MINECO). M.J.B. is supported by the Agencia Estatal de Investigación project PID2021-123321OB-I00 funded by MCIN /AEI /10.13039/501100011033/ FEDER, UE; the Gilead Fellowship GLD22/00152, and the Miguel Servet program funded by the Spanish Health Institute Carlos III (CPII22/00005). N.M.C. was supported by S2022/BMD-7209 (INTEGRAMUNE-CM) to N.M.C. C.M.C. was supported by FIS.18/01163. F.S.M. and N.M.C. were also supported by La Caixa Health Research Grant LCF/PR/HR23/52430018. Grants to A.A. from the Fondo de Investigación Sanitaria del Instituto de Salud Carlos III, co-funded by the Fondo Europeo de Desarrollo Regional (FEDER) (FIS PI19/00549; FIS PI22/01542), and Sociedad Cooperativa de Viviendas Buen Suceso, S. Coop. Mad; to A.A. and F.S.M. from the Fondo de Investigación Sanitaria del Instituto de Salud Carlos III, co-funded by the Fondo Europeo de Desarrollo Regional (FEDER) (CIBER Cardiovascular). A.A. was supported by CIBER cardiovascular (CIBERCV) from Instituto de Salud Carlos III. L.E.P. was financed by Inmunovacter REACT-UE (Comunidad de Madrid). We also would like to thank Verónica Labrador from the Microscopy Core from Centro Nacional de Investigaciones Cardiovasculares for technical support in confocal image analysis.

## Author contributions

E.M.G, I.T., I.S.C., F.S.M., J.A., and A.A. developed the research idea and study concept, designed the study and wrote the manuscript. E.M.G. supervised the study. F.S.M., A.A and M.J.B. also provided feedback during the study. I.T. and I.S.C. conducted most of the experiments and data analysis. O.R., J.A., E.A., and G.I. provided PBMC and BAL samples from COVID-19 patients. F.S.M., I.G.A., C.M.C., M.J.C. and I.S. provided additional COVID-19 PBMC samples and clinical information. O.P., M.C.M., and C.D.A. contributed to analysis of Mo subsets in BAL and PBMC cultures from COVID-19 patients. A.M. and C.M.C. contributed to the collection of PBMC from COVID-19 patients. P.M.F. performed the multiparametric analyses of clinical and immunological parameters in BAL samples. C.S. participated in the immunofluorescence confocal microscopy image collection. M.C.M. contributed to statistical analysis of correlation networks and confocal microscopy image analysis. M.J.B. and M.G. provided reagents and critical feedback for innate and adaptive immune responses assays and on tissue memory T cells and cytokine profiles. F.S.M., N.B.M.C. and M.J.C. provided antibodies and critical feedback for Western Blot analyses. J.P.C. provided lung tissue samples from critical COVID-19 patients used for histology analysis. A.A. and L.E. provided SARS-CoV-2 peptides and contributed to T cell assay optimization. E.M.G. is a senior author supervised the study.

## Competing interests

I.G.A. reports the following competing interests: personal fees from Lilly and Sanofi; personal fees and non-financial support from BMS; personal fees and non-financial support from Abbvie; research support, personal fees and non-financial support from GEBRO Pharma; non-financial support from MSD, Pfizer and Novartis, not related to the submitted work. The rest of the authors have no additional competing interests.

## Additional information

[1] Medicine Faculty, Universidad Autónoma de Madrid, Madrid, Spain. [2] Immunology Unit from Hospital Universitario La Princesa, Instituto Investigación Sanitaria-Princesa IIS-IP, Madrid, Spain. [3] CIBER Infectious Diseases (CIBERINFECC), Instituto de Salud Carlos III, Madrid, Spain. [4] Pneumology Unit from Hospital Universitario La Princesa, Madrid, Spain. [5] Infectious Diseases Department, Institut de Recerca Hospital Universitari Vall d'Hebrón (VHIR), Universitat Autònoma de Barcelona, Barcelona, Spain. [6] Infectious Diseases Unit from Hospital Universitario La Princesa, Madrid, Spain. [7] Centro de Investigación Biomédica en Red de Enfermedades Respiratorias (CIBERES), Instituto de Salud Carlos III, Madrid, Spain. [8] Rheumatology Department from Hospital Universitario La Princesa. Instituto de Investigación Sanitaria-Princesa IIS-IP, Madrid, Spain. [9] Department of Pathology, Hospital Universitario Ramón y Cajal. Instituto Ramón y Cajal de Investigaciones Sanitarias (IRYCIS), Universidad de Alcalá. Centro de Investigación Biomédica en Red de Cáncer (CIBERONC), Madrid, Spain. [10] CIBER Cardiovascular, Instituto de Salud Carlos III, Madrid, Spain. [11] These authors contributed equally: Ilya Tsukalov, Ildefonso Sánchez-Cerrillo. ✉e-mail: enrique.martin@uam.es

