## [Peer Review File · Nature Communications]

NF κ B and NLRP3/NLRC4 inflammasomes regulate differentiation, activation and functional properties of monocytes in response to distinct SARS-CoV-2 proteinsREVIEWER COMMENTS

Reviewer #1 (Remarks to the Author):

The current manuscript describes the effects of SARS CoV2 proteins S1 and NP on the phenotype, activity, and interactions of Monocytes subpopulations, looking at the effects of the above proteins on NF-kB and inflammasome activity and expression in the different monocyte subpopulations. The team of authors discovered that S1 and NP behave differently in the different Mo subpopulations and trigger different responses, thus describing novel results.

However, it is not clear the significance of these findings, since activation of NLRP3 and NFkB in SARSCoV2 has been previously reported, and NLRP3 inhibitors or IL-1 blockers finished or are completing clinical testing in patients with COVID-19.

The study design is based on the use of PBMCs from healthy and COVID-19 patients and on cells from BAL of COVID-19 patients. A control BAL would have made a better control than PBMCs, although it is understandable of being difficult to collect such samples.

As the authors report as a limitation, there is a lack of data obtained with additional cell types in the blood and lungs, as cell-cell interactions may modify the observed results in the presence of S1 and NP. Several results are pure observations, without an in depth look at the mechanism of action or determination of in vivo relevance of the findings (e.g. how does a modification of NLRP3 or NLRC4 in a Mo subpopulation affect the course of disease?)

Figure 1 and other figures: Why the number of samples analyzed differs from the number of patients enrolled? And why were the missing samples not included in the analysis?

Also some WB images (e.g Figure 2C) and respective bar-graph do not represent a similar trend (see active caspase in S1 compared to the others).

In addition, in figure 2AC, to prove an inflammasome mediate activation of caspase-1, NLRP3 and NLRC4 inhibitors or siRNA should be used to show that those proteins mediate activation of caspase-1. Similarly, a similar approach should be used in the other assays where IL-1b and other cytokines protein levels are measured, to define whether with and without inflammasome receptor inhibition the amount of active protein secreted changes. This is done with S1 but not with NP. In addition, S1 induction of IL-1b is not abrogated by the pan caspase inhibitor, which suggests that a wrong concentration or some other technical problem may have occurred. As additional comment, I would not expect such low levels of active IL-1b (1 to 4 pg/ml) in cultured cells if the inflammasome is activated. I wonder if the IL-1 measured is the result of passive release due to cell death and not active production and release through N-ter gasdermin D pores. This plus the combination of use of the IL-1b ELISA assay from R&D in this work should measure both pro and active IL-1b. Such low levels of IL-1beta may be due to the lack of inflammasome priming. In vivo, a viral infection (and not just a purified protein like S1 or NP) with the induction of cell death or cytokine release promotes inflammasome priming. Therefore, to show that S1 and NP are good inflammasome activators, the addition of a priming phase would give a clear signal (high IL-1 b secretion).

Lastly, in the ELISA assay results, the values are reported as fold change but measured as pg/ml. It is confusing. The results as absolute pg/ml values should be reported as well.

Figure S2. There are some groups that have fewer samples compared to the others. Why

were some samples excluded?

Reviewer #2 (Remarks to the Author):

This is an interesting a well-written study on the role of the activation of NLRP3 inflammasome in critical COVID-19 both for the production of IL-1 β but also for the differentiation of T-cells. I have some concerns for this submission.

- Why was not pro-IL-1 β measured in the cell cytosols?
- NF-kB does not mediate only the production of pro-IL-1 β but also of other cytokines. In this aspect, TNF and IL-18 measurements are missing. IL-18 is crucial since it follows the kinetics of IL-1 β .
- The authors need to better address how the transformation of monocytes into dendritic cells mediate the T-cell activation. IL-17 and IFN γ measurements are not enough since they may come from monocytes themselves or $\gamma\delta$ T-cells.
- The clinical parameters presented in Figure 6 have nothing to do with the activation of NLRP3 of COVID-19. Hyper-production of IL-1 β is associated with ferritin, fibrinogen and triglycerides. The authors need to associate their findings with these variables and the components of macrophage activation.
- It is a real surprise that the authors do not mention anything on the approval of anakinra for COVID-19 patients at risk through the early activation of the IL-1 pathway detected by the biomarker suPAR. The drug is licensed by both the EMA and FDA. A discussion on this strengthens the findings of the authors.

REVIEWER COMMENTS

Reviewer #1 (Remarks to the Author):

The current manuscript describes the effects of SARS CoV2 proteins S1 and NP on the phenotype, activity, and interactions of Monocytes subpopulations, looking at the effects of the above proteins on NF-kB and inflammasome activity and expression in the different monocyte subpopulations. The team of authors discovered that S1 and NP behave differently in the different Mo subpopulations and trigger different responses, thus describing novel results. We thank the reviewer for his/her appraisal of the novel findings regarding the innate sensing of different SARS-CoV-2 proteins by monocytes of our manuscript.

However, it is not clear the significance of these findings, since activation of NLRP3 and NFkB in SARSCoV2 has been previously reported, and NLRP3 inhibitors or IL-1 blockers finished or are completing clinical testing in patients with COVID-19.

Thanks for bringing up this point. The novelty of our study relies on the identification of non-redundant roles of inflammasome versus NFkB in regulating Monocyte differentiation and activation in response to SARS-CoV-2 proteins as independent mechanisms. Moreover, we also have identified different inflammasome sensors that may play a different role in the detection of specific viral proteins. Finally, we are also providing associations between the induction of these specific innate responses and functional abilities of Mo mediating activation of CD4/CD8 T cells *in vitro*. All these aspects and their potential relevance have been highlighted in the main manuscript and discussion of our revised manuscript.

The study design is based on the use of PBMCs from healthy and COVID-19 patients and on cells from BAL of COVID-19 patients. A control BAL would have made a better control than PBMCs, although it is understandable of being difficult to collect such samples.

To address the reviewer's request, we have now included an additional set of data set of BAL infiltrates from non-COVID-19 patients suffering lung cancer or interstitial disease recruited from our hospital, where we also show that stimulation with SARS-CoV-2 S1 peptides leads to an increase in the proportions of NC Mo.

As the authors report as a limitation, there is a lack of data obtained with additional cell types in the blood and lungs, as cell-cell interactions may modify the observed results in the presence of S1 and NP. Several results are pure observations, without an in depth look at the mechanism of action or determination of in vivo relevance of the findings (e.g. how does a modification of NLRP3 or NLRC4 in a Mo subpopulation affect the course of disease?)

We appreciate the comment made by the reviewer. We agree that the participation of additional cell types present in the blood and lung cannot be ruled out, and therefore we have highlighted this limitation in the discussion of our revised manuscript. In addition, to address the concern of lack of connection between the activation of the inflammasome and the in vivo/functional relevance, we are now providing a new data set of histologic analysis of caspase 1/inflammasome activation on monocytes infiltrated in lung from severe COVID-19 patients. We have also analysed the presence of activated IL-17+/IFN \$\gamma\$ + T cells in **highly infiltrated tissue areas**. Moreover, we have further extended our analysis of the implication of NLRP3/NLRC4 inflammasomes by directly studying the

impact of siRNA mediated knock down of these sensors in the activation of IFN γ + / Th17+ CD4+ T cell and IFN γ + CD8+ T cell responses *in vitro*.

Figure 1 and other figures: Why the number of samples analyzed differs from the number of patients enrolled? And why were the missing samples not included in the analysis?

Thanks for bringing up this point. We have now highlighted that the total number of samples recruited from our study cohorts were not all used for each experimental dataset in the methods section. We have further highlighted the N from each cohort used for each individual experiment in the methods and figure legends of our revised manuscript.

Also some WB images (e.g Figure 2C) and respective bar-graph do not represent a similar trend (see active caspase in S1 compared to the others). We have now increased the N and included a new representative WB gel image where the tendencies of the plots are more evident and some the differences in response to S1 are now significant for caspase-1 (see revised Figure 2).

In addition, in figure 2AC, to prove an inflammasome mediate activation of caspase-1, NLRP3 and NLRC4 inhibitors or siRNA should be used to show that those proteins mediate activation of caspase-1. Similarly, a similar approach should be used in the other assays where IL-1 β and other cytokines protein levels are measured, to define whether with and without inflammasome receptor inhibition the amount of active protein secreted changes. This is done with S1 but not with NP.

To address the direct involvement of NLRP3 and NLRC4 in Caspase 1 activation in response to S1 and NP respectively, we are now providing new confocal microscopy images where we show that Caspase 1 and NLRP3 or NLRC4 co-localize after exposure to S1. Also, we have analysed whether siRNA mediated-knock down of these sensors affects IL1- β secretion in culture supernatants of Mo treated either with Scramble or NLRP3 or NLRC4-specific siRNAs using ELISA results (New supplemental Figure 3F). Finally, we also have addressed the impact of NLRP3 and NLRC4 specific siRNAs in the functional abilities of Mo to induce IFN γ + and IL-17+ CD4+ T cells and IFN γ + and CD17a+ CD8+ T cells (See new figure 6 and new figure 6).

In addition, S1 induction of IL-1 β is not abrogated by the pan caspase inhibitor, which suggests that a wrong concentration or some other technical problem may have occurred.

We thank the reviewer for noticing this error. Caspase inhibitors were mislabelled in our original Supplemental Figure 3A and the pan caspase inhibitor (Z-FMK-VAD) was indeed the most effective reducing the secretion IL1, in contrast to a partial inhibitor (Z-FMK-WEHD), we have now corrected the labelling and also provided raw instead to normalized data.

As additional comment, I would not expect such low levels of active IL-1b (1 to 4 pg/ml) in cultured cells if the inflammasome is activated. I wonder if the IL-1 measured is the result of passive release due to cell death and not active production and release through N-ter gasdermin D pores. This plus the combination of use of the IL-1b ELISA assay from R&D in this work should measure both pro and active IL-1b. Such low levels of IL-1beta may be due to the lack of inflammasome priming. In vivo, a viral infection (and not just a purified protein like S1 or NP) with the induction of cell death or cytokine release promotes inflammasome priming. Therefore, to show that S1 and NP are good inflammasome activators, the addition of a priming phase would give a clear signal (high IL-1 b secretion). Lastly, in the ELISA assay results, the values are reported as fold change but measured as pg/ml. It is confusing. The results as absolute pg/ml values should be reported as well.

We thank the reviewer for bringing up this excellent point. We are providing additional data showing that mild but significant differences were observed in cell viability are observed between Mo exposed to DMSO or to S1 peptides (from 67% to 52% of viable cells), which suggests the increase of IL-1 is not driven by a massive cell death mediated by Gasdermin D. We have also included western blot analysis of proteolytic cleavage of Gasdermin D in the supplemental material of the revised manuscript. In addition, the original plot represented the fold change in IL1beta concentrations measured, but we agree the axis labelling is misleading and did not represent 4 pg/mL, but 4 times higher concentration in pg/mL. We actually are detecting in the order of 20-100pg/mL of IL1beta in our assays, which should represent active-form of IL-1b. To avoid confusion and to address that same concern raised by the reviewer we are now providing the raw data of these analyses in our revised manuscript. Also, NLRP3 inflammasome has also been reported to induce gasdermin D independent secretion of IL1beta (PMID: 34678046), and we have discussed this study in our revised manuscript.

Figure S2. There are some groups that have fewer samples compared to the others. Why were some samples excluded?

As the reviewer pointed out, in the RT-qPCR analyses from cultured Mo shown in Supp Fig2, we not always had enough cells to include all the conditions, or material to analyse all transcripts included. For this reason, we have mismatching number of samples for some of these experiments.

Reviewer #2 (Remarks to the Author):

This is an interesting a well-written study on the role of the activation of NLRP3 inflammasome in critical COVID-19 both for the production of IL-1 β but also for the differentiation of T-cells.

First of all, we would like to thank the reviewer for acknowledging the interest of our study.

I have some concerns for this submission.

- Why was not pro-IL-1 β measured in the cell cytosols?

We have now included a new set of flow cytometry data showing total levels intracellular IL-1 β (containing Pro-IL1 β) in the cytoplasm of Mo exposed to SARS-CoV-2 proteins or DMSO. We have also included analysis of levels of pro-IL1 and active IL-1 β by western blot in the supplemental material from our revised manuscript.

- NF- κ B does not mediate only the production of pro-IL-1 β but also of other cytokines. In this aspect, TNF and IL-18 measurements are missing. IL-18 is crucial since it follows the kinetics of IL-1 β .

Thanks for the excellent suggestion, we have now included RT-qPCR results of TNF α and IL-18 transcripts in our revised manuscript. We have found that similarly to IL-8, TNF α is also induced in Mo after stimulation with S1, while expression of IL-18 was more variable and we did not find consistent significant changes in transcription of this cytokines.

- The authors need to better address how the transformation of monocytes into dendritic cells mediate the T-cell activation. IL-17 and IFN γ measurements are not enough since they may come from monocytes themselves or $\gamma\delta$ T-cells.

We thank the reviewer for the comments. In our study, we used freshly isolated Mo exposed to viral proteins for the MLR assays and analyzed the allogeneic T cell response after 3 days, so we believe it is unlikely that these cells have differentiated into DCs, since no previous culture in the presence of IL-4 and GM-CSF were performed and a short culture was performed prior the regular times needed for the potential differentiation of Mo to DC (typically from 5-7 days). In addition, we analyzed expression of IL-17 and IFN γ where CD3 $^+$ CD8 $^-$ and/or CD3 $^+$ CD8 $^+$ T cells from Peripheral blood used for MLR which may represent lymphocytes which are in their majority TCR alpha/beta $^+$ cells, which in an MLR assays will be the main cells responding to the Mo via MHC-II/I. However, in the analyses of IL-17 expression using BAL from COVID-19 we cannot discard that we may have gamma delta T cells in our cultures contributing to this response, since previous studies suggest that these cells may be altered/present in COVID-19 patients. We have included these limitations in the discussion of our revised manuscript.

- The clinical parameters presented in Figure 6 have nothing to do with the activation of NLRP3 of COVID-19. Hyper-production of IL-1 β is associated with ferritin, fibrinogen and triglycerides. The authors need to associate their findings with these variables and the components of macrophage activation.

We thank the reviewer for the valuable feedback. In addition to clinical parameters previously associated with inflammation in severe COVID-19 identified by our and other previous studies, we have now included the suggested ferritin, fibrinogen and triglycerides values as a surrogate of IL-1 β and inflammasome activity.

- It is a real surprise that the authors do not mention anything on the approval of anakinra for COVID-19 patients at risk through the early activation of the IL-1 pathway detected by the biomarker suPAR. The drug is licensed by both the EMA and FDA. A discussion on this strengthens the findings of the authors.

We thank the reviewer for the excellent comment. As requested, we have now further discussed the impact of anakinra targeting the IL-1beta pathway and the impact on suPAR in our revised manuscript.

REVIEWER COMMENTS

Reviewer #1 (Remarks to the Author):

This reviewer appreciates the effort put in by the authors to perform additional experiments to validate and better support the findings. Despite the overall improvement of the data, some of the new experiments present few inconsistencies or do not support the conclusions.

Additional concerns:

- 1) Figure 3 C: the top and bottom row report NLRP3. A ratio of NLRP3+ CD14 vs NLRC4+ CD14 cells would be nice data to add. As a reviewer, I understand this data would not add much to the message. Therefore, it is not necessary data but would be added information on inflammasome formation in vivo that can be quoted in future manuscripts.
- 2) WBs in Supplemental Figure 2 lack the GAPDH. GAPDH bands would help to better judge the data.
- 3) In supplemental figure 3, the inefficacy of the specific caspase-1/4/5 inhibitor WEHD indicates some technical problems.
- 4) In Supplemental Figure 3 F, the raw data show that the S1 and NP in the sc siRNA group do not produce any effect.
- 5) In the discussion, Anakinra is described as an anti-IL1R, a terminology used for antibodies, when in reality, this is not an antibody but an IL1R antagonist.
- 6) The significance of the findings still needs to be put into perspective with the fact that NLRP3 and NFkB in SARSCoV2 have been previously reported, and NLRP3 inhibitors or IL-1 blockers finished or are completing clinical testing in patients with COVID-19. The fact that IL-1Ra is already approved for use in patients with COVID-19 in the US and Europe is not discussed, and the overall relevance of the data is unclear.

Reviewer #2 (Remarks to the Author):

The manuscript is substantially improved. The only concern is that the reference supporting the registration of Anakinra is wrong.

REVIEWER COMMENTS

Reviewer #1 (Remarks to the Author):

This reviewer appreciates the effort put in by the authors to perform additional experiments to validate and better support the findings. Despite the overall improvement of the data, some of the new experiments present few inconsistencies or do not support the conclusions.

Additional concerns:

1) Figure 3 C: the top and bottom row report NLRP3. A ratio of NLRP3+ CD14 vs NLRC4+ CD14 cells would be nice data to add. As a reviewer, I understand this data would not add much to the message. Therefore, it is not necessary data but would be added information on inflammasome formation in vivo that can be quoted in future manuscripts.

We have corrected the mislabeling of the bottom images on Figure 3C which corresponds to NLRC4 as specified in the legend. As requested, we have also provided a quantification of NLRP3+CD14+cells/total CD14+ and NLRC4+CD14+/CD14+ as well as cells co-expressing also caspase-1 in COVID-19 lung sections.

2) WBs in Supplemental Figure 2 lack the GAPDH. GAPDH bands would help to better judge the data.

We have now included the GAPDH data from the WB in Supplemental Figure 2.

3) In supplemental figure 3, the inefficacy of the specific caspase-1/4/5 inhibitor WEHD indicates some technical problems.

We agree that the WEHD was more inefficient in our assays reducing the IL-1 β secretion in culture supernatants than the other inhibitors. For this reason, we have removed the data from this inhibitor from the main Figure 3 and supplemental Figure 3 and we have kept the data from the other two inhibitors (MCC950 NLRP3 inhibitor, Z-VAD-FMK panCaspase-1 inhibitor) for which we confirmed the ability to significantly reduce IL-1 β secretion, which strongly supports that inhibition of the inflammasome activation is not required to Mo differentiation.

4) In Supplemental Figure 3 F, the raw data show that the S1 and NP in the sc siRNA group do not produce any effect.

The raw levels for IL1 β detection were variable in our assays and that is why we normalized the data to the unstimulated condition. However, we showed the original data for transparency reasons, how it was required by reviewers. In addition, we have now included a paired representation of these raw data that was originally presented in the Supplemental figure3G showing that there is an induction in NP SC compared to DMSO SC condition in 4 out of 6 tested donors, despite the variability.

5) In the discussion, Anakinra is described as an anti-IL1R, a terminology used for antibodies, when in reality, this is not an antibody but an IL1R antagonist.

We apologize for the mistake, we have now corrected and specified in the discussion that anakinra is an IL1R antagonist.

6) The significance of the findings still needs to be put into perspective with the fact that NLRP3 and NF κ B in SARSCoV2 have been previously reported, and NLRP3 inhibitors or IL-1 blockers finished or are completing clinical testing in patients with COVID-19. The fact that IL-1Ra is already approved for use in patients with COVID-19 in the US and Europe is not discussed, and the overall relevance of the data is unclear.

We have now further highlighted the fact that activation of the NLRP3 and NF κ B pathways had been described in the context of SARS-CoV2 infection and we have also stressed how these pathways differentially affect Mo differentiation versus activation and this is one of the novel findings of our study. In addition, we have also further discussed that anakinra has also been approved for COVID-19 in the US and Europe in our revised manuscript.

Reviewer #2 (Remarks to the Author):

The manuscript is substantially improved. The only concern is that the reference supporting the registration of Anakinra is wrong.

We apologize, we have now provided the correct references for anakinra registration and efficacy trial in COVID-19.

REVIEWERS' COMMENTS

Reviewer #1 (Remarks to the Author):

The authors replied to all my previous queries. I have no other comments for the authors. I appreciated all the effort they have made to reply to my comments and improve clarity of the MS.